# Learning Diffusion Models with Flexible Representation Guidance

Chenyu Wang[1*]   Cai Zhou[1*]   Sharut Gupta[1]   Zongyu Lin[2]
Stefanie Jegelka[13]   Stephen Bates[1]   Tommi Jaakkola[1]
[1]MIT  [2]UCLA  [3]TU Munich

 github.com/ChenyuWang-Monica/REED    Project Page

## Abstract

Diffusion models can be improved with additional guidance towards more effective representations of input. Indeed, prior empirical work has already shown that aligning internal representations of the diffusion model with those of pre-trained models improves generation quality. In this paper, we present a systematic framework for incorporating representation guidance into diffusion models. We provide alternative decompositions of denoising models along with their associated training criteria, where the decompositions determine when and how the auxiliary representations are incorporated. Guided by our theoretical insights, we introduce two new strategies for enhancing representation alignment in diffusion models. First, we pair examples with target representations either derived from themselves or arisen from different synthetic modalities, and subsequently learn a joint model over the multimodal pairs. Second, we design an optimal training curriculum that balances representation learning and data generation. Our experiments across image, protein sequence, and molecule generation tasks demonstrate superior performance as well as accelerated training. In particular, on the class-conditional ImageNet $256 \times 256$ benchmark, our guidance results in 23.3 times faster training than the original SiT-XL as well as four times speedup over the state-of-the-art method REPA [61].

## 1   Introduction

Diffusion models [20, 50] have achieved significant empirical success across images, videos, audio, and biomolecules [46, 6, 44, 37, 1, 56]. While their primary objective is to model the data distribution, recent work [10, 32, 58] show that diffusion models implicitly learn some discriminative features in their latent representations. Nevertheless,

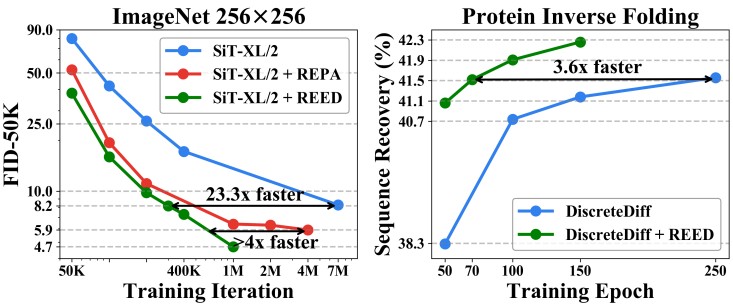

Figure 1: REED achieves superior performance and accelerated training on image generation and protein inverse folding.

the learned representations still fall behind those of pretrained representation learning models, which potentially limits the performance of generative modeling. Consequently, integrating high-quality representations obtained from other pretraining tasks such as Self-supervised Learning (SSL) unlocks additional possibilities to improve diffusion training. Representations can be integrated through flexible designs, for example, via conditioning [34, 35] or alignment [61]. Specifically, RCG [34]

*Equal Contribution: {wangchy, caiz428}@mit.edu

trains a generative model to produce semantic representations from a self-supervised encoder and conditions image generation on these representations, while REPA [61] aligns diffusion model hidden states with clean representations from pretrained encoders. Despite the empirical effectiveness of these methods, there is a lack of systematic understanding of the role of representation learning in training diffusion models.

In this paper, we present a theoretical framework to analyze how high-quality, pretrained representations can enhance diffusion model training. Building on the DDPM [20] framework, we derive a variational bound on the negative log-likelihood that incorporates the pretrained representations to guide the generation through a probabilistic decomposition of the joint distribution and a weighted aggregation process. This leads to two key components: *representation modeling*, which promotes extracting representations from noisy input, and *data generation*, which fuses representation-conditioned and unconditional generation into a hybrid model governed by a weight schedule (Section 2.1). The framework generalizes to multi-latent structures, enabling integration of diverse representations (Section 2.2). Our unified perspective subsumes existing approaches like REPA [61] and RCG [34], offering theoretical insight into representation-enhanced diffusion models. In particular, REPA emerges as a special case with a linear weight aggregation schedule and a single latent, where the output of the hybrid model is approximated by conditioning solely on representations extracted from noisy inputs to mitigate the training-inference gap (Section 2.3). We also provide a TV distance error bound for the sample distribution of representation-aligned diffusion models (Appendix C.2).

Built on the theoretical insights, our general framework allows for a much broader design space for exploiting representation alignment [61] in diffusion models. In particular, we demonstrate the effectiveness of two novel strategies, designated as **REED** (**R**epresentation-**E**nhanced **E**lucidation of **D**iffusion). First, we argue that diffusion models can profit from diverse representation sources, hence we incorporate multi-latent representations from different modalities to provide complementary information. Our new multimodal alignment scheme uses synthetic paired data and multimodal pretrained models trained on datasets from different corpora using discriminative or self-supervised tasks to generate the target representations (Section 3.1). Second, we analyze optimal training curricula for representation-aligned diffusion models, demonstrating that representation learning is crucial in the early training stage to yield substantial advantages. Consequently, we introduce a novel curriculum that applies the representation learning loss from the outset, while progressively increasing the diffusion loss coefficient from zero to its maximum value as a phase-in protocol (Section 3.2). Finally, we propose three practical instantiations of our framework that yield improvements in image, protein, and molecule generation (Section 3.3).

We extensively evaluate our proposed REED across image generation, protein inverse folding, and molecule generation tasks. For the class-conditional ImageNet $256 \times 256$ benchmark, aligning SiT-XL/2 with DINOv2 [42] and Qwen2-VL [55] representations using our improved curriculum significantly improves the generation quality measured by FID scores. REED achieves a $23.3\times$ training speedup over the original SiT-XL, reaching FID=8.2 in only 300K training iterations (without classifier-free guidance [19]); and a $4\times$ speedup over REPA [61], matching its classifier-free guidance performance at 800 epochs with only 200 epochs of training (FID=1.80). For protein inverse folding, aligning discrete diffusion models trained on the PDB dataset with AlphaFold3 [1] structure and sequence representations accelerates training by $3.6\times$ and yields significantly superior performance across metrics such as sequence recovery rate, RMSD and pLDDT. For molecule generation, aligning state-of-the-art geometric equivariant flow matching (SemlaFlow [25]) with pretrained Unimol [64] representations improves metrics such as atom and molecule stability, validity, energy, and strain on the challenging Geom-Drug [3] datasets.

We summarize our main contributions as follows:

- We conduct rigorous analysis for representation-enhanced diffusion model training, delivering theoretical insights into prior empirical methods and informing the design of new strategies.

- We systematically investigate the design space for representation guidance in training diffusion models, proposing a novel multimodal representation alignment scheme via synthesized paired multimodal data, an effective training curriculum, and practical domain-specific instantiations.

- Extensive experiments demonstrate that our method consistently accelerates training and improves performance across various diffusion model types and diverse tasks, including image, protein sequence, and molecule generation tasks.

## 2 Theoretical Characterizations

In this section, we introduce a general theoretical framework for representation-enhanced diffusion models. Building on the DDPM framework [20], we explore how representations can offer additional information and enhance the generation process. Notably, due to the known equivalence between DDPM, score matching [50], stochastic interpolants [2], and flow matching [38], our insights are applicable to a broad class of diffusion model variants. Proofs for this section are detailed in Appendix C.

### 2.1 Variational Bound with Decomposed Probabilistic Framework

**Problem Formulation.** We consider a generative modeling setting involving discrete-time latent variables $x_{0:T}$ and an auxiliary representation $z$, structured according to the Markov chain:

$$x_T - x_{T-1} - ... - x_1 - x_0 - z \tag{1}$$

The data distribution follows the factorization $q(x_{0:T}, z) = q(x_0)q(x_{1:T}|x_0)q(z|x_0)$, where $q(x_0)$ denotes the empirical data distribution, $q(x_{1:T}|x_0)$ corresponds to the forward diffusion process, and $q(z|x_0)$ represents the mapping of the data to a semantic latent space through a pretrained encoder.

Our goal is to learn a generative model $p_\theta(x_0)$, defined implicitly through a joint model $p_\theta(x_{0:T}, z)$, such that $p_\theta(x_0) := \int p_\theta(x_{0:T}, z)dx_{1:T}dz$. Following the variational framework used in diffusion-based generative models [20, 48] (see Appendix B for details), we derive a variational bound on the negative log-likelihood of $p_\theta(x_0)$ that incorporates the latent representation $z$ to guide the generation process. The bound $\mathcal{L}_{VB}^z$ is formalized in Equation (2):

$$\mathcal{L}_{VB}^z := \mathbb{E}_{q(x_{0:T}, z)}\left[\log \frac{q(z|x_0) \cdot \prod_{t=1}^T q(x_{t-1}|x_t, x_0) \cdot q(x_T|x_0)}{p_\theta(x_{0:T}, z)}\right] \geq -\mathbb{E}_{q(x_0)} \log p_\theta(x_0) \tag{2}$$

**Decomposition of Model Parameterization.** The denominator of $\mathcal{L}_{VB}^z$, $p_\theta(x_{0:T}, z)$, defines how the latent variable model is parameterized for the reverse diffusion process. The way in which $z$ is integrated into the reverse diffusion steps–specifically the inference of $x_{t-1}$ from a noisier state $x_t$–is crucial for representation-enhanced diffusion models. When $z$ is introduced earlier in the process (at larger values of $t$), the model benefits from more accurate estimation via $p_\theta(x_{t-1}|x_t, z)$ rather than $p_\theta(x_{t-1}|x_t)$, since conditioning on $z$ leads to a better approximation of the posterior $q(x_{t-1}|x_t, x_0)$. However, this comes at the cost of a less accurate estimation with $p_\theta(z|x_t)$, as the dependency path between $x_t$ and $z$ becomes longer based on the Markov structure, making it harder to approximate the true posterior $q(z|x_0)$. This is formally reflected in the decomposition of joint probability $p(x_{0:T}, z)$:

$$p(x_{0:T}, z) = p(z|x_t) \cdot \prod_{i=1}^t p(x_{i-1}|x_i, z) \cdot \prod_{i=t+1}^T p(x_{i-1}|x_i) \cdot p(x_T) := p^t(x_{0:T}, z), \ \forall t = 0, 1, ..., T^2 \tag{3}$$

*Remark* 1. Each decomposition above is an exact equality for any $t$. The optimal components $p(x_{i-1}|x_i, z)$ and $p(x_{i-1}|x_i)$ matching the forward process $q(x_{0:T}, z)$ are consistent across all $t$s. Thus, identical components can be used in every decomposition without loss of accuracy.

*Remark* 2. At any timestep $t$, the marginal distribution of $x_t$ remains invariant across all decompositions. Consequently, $z$ can be resampled at any step while preserving the same marginal distribution.

The above remarks indicate that representation-guided decompositions can be arbitrarily combined without incurring additional loss. To control this combination, we introduce a novel timestep weighting schedule $\{\alpha_t\}_{t=1}^T$, where $\alpha_t \geq 0$ are hyperparameters with $\sum_{t=1}^T \alpha_t = 1$. This produces an equivalent formula of the joint distribution, i.e., $p_\theta(x_{0:T}, z) = \sum_{t=1}^T \alpha_t p_\theta^t(x_{0:T}, z)$, which allows us to adjust when and how the representation $z$ is injected into the generative process. Such formula results in a hybrid conditional distribution $\tilde{p}_\theta(x_{t-1}|x_t, z; A_t)$ (Equation (5)) governed by the cumulative weights $A_t := \sum_{i=t}^T \alpha_i$. $A_t$ decreases with $t$ and serves as a schedule for gradually introducing guidance from $z$ by interpolating between guided and unguided reverse diffusion. By choosing $\alpha_t$, and thus $A_t$, we can flexibly control the influence of $z$ throughout the diffusion process. The formal expression for the resulting $\mathcal{L}_{VB}^z$ is given below.

---

[^2]: When $t = 0$ or $t = T$, the corresponding products reduce to 1.

**Proposition 1** (Decomposition Structure of the Variational Bound). *Let $\{\alpha_t \geq 0\}_{t=1}^T$ be a set of weights summing to one, and define $A_t \in [0,1] := \sum_{i=t}^T \alpha_i$. Then, the variational bound $\mathcal{L}_{VB}^z$ in Equation (2) can be written as:*

$$\mathcal{L}_{VB}^z = \sum_{t=1}^T \mathbb{E}_{q(x_{0:T},z)}\left[\log \frac{q(x_{t-1}|x_t,x_0)}{\tilde{p}_\theta(x_{t-1}|x_t,z;A_t)}\right] - \sum_{t=1}^T \mathbb{E}_{q(x_{0:T},z)}\left[\log Z_t(x_t,z;A_t)\right]$$

$$+ \sum_{t=1}^T \alpha_t \mathbb{E}_{q(x_{0:T})}\left[D_{KL}(q(z|x_0)||p_\theta(z|x_t))\right] \tag{4}$$

*where $\tilde{p}_\theta(x_{t-1}|x_t,z;A_t)$ and its normalization $Z_t(x_t,z;A_t)$ are defined as:*

$$\tilde{p}_\theta(x_{t-1}|x_t,z;A_t) = \frac{1}{Z(x_t,z)} p_\theta^{A_t}(x_{t-1}|x_t,z) p_\theta^{1-A_t}(x_{t-1}|x_t) \tag{5}$$

$$Z_t(x_t,z;A_t) = \int p_\theta^{A_t}(x_{t-1}|x_t,z) p_\theta^{1-A_t}(x_{t-1}|x_t) dx_{t-1} \tag{6}$$

The first term in Equation (4) corresponds to a KL divergence between the data distribution $q(x_{t-1}|x_t,x_0)$ and the hybrid model estimation $\tilde{p}_\theta(x_{t-1}|x_t,z;A_t)$, serving as the primary data generation objective. The second term, involving the expectation of log-normalization constants, can be calculated as the discrepancy between conditional and unconditional model predictions, multiplied by the coefficient $A_t(1-A_t)$ (see Appendix C.1.4 for details). The third term is the KL divergence between the predicted $p_\theta(z|x_t)$ and the reference distribution $q(z|x_0)$. Optimizing this term drives $p_\theta(z|x_t)$ to better approximate the true (but intractable) $q(z|x_t)$, complementing the second term. Together, these terms facilitate the representation learning procedure where the model uses $x_t$ to recover meaningful semantic information about $z$.

## 2.2 Multi-Latent Representation Structure

The expressiveness of latent representations can be significantly enhanced through a multi-latent structure, as shown in prior work such as NVAE [52]. Instead of modeling a single representation $z$, we extend our formulation to a series of representations $\{z_l\}_{l=1}^L$, each indicating a different level of abstraction. This leads to a similar Markov chain: $x_T - x_{T-1} - ... - x_1 - x_0 - \mathbf{z}^L$, with $\mathbf{z}^L = \{z_l\}_{l=1}^L$ representing the collection of representations from pretrained models, $\mathbf{z}^L \sim q(\mathbf{z}^L|x_0)$. Thus, the variational bound $\mathcal{L}_{VB}^z$ extends naturally to $\mathcal{L}_{VB}^{\mathbf{z}^L}$, where all instances of $z$ are replaced with $\mathbf{z}^L$. Notably, the generation process of $\mathbf{z}^L$ in the third KL divergence term of $\mathcal{L}_{VB}^{\mathbf{z}^L}$ can be realized with a sequential structure, where each representation level $z_l$ depends on both the noisy input $x_t$ and all previous latents $z_{<l}$[3]:

$$D_{KL}(q(\mathbf{z}^L|x_0)||p_\theta(\mathbf{z}^L|x_t)) = \sum_{l=1}^L \mathbb{E}_{q(z_{<l}|x_0)}\left[D_{KL}(q(z_l|x_0)||p_\theta(z_l|x_t,z_{<l}))\right] \tag{7}$$

This multi-latent formulation enables the model to leverage semantic features across different granularity. Previous hierarchical VAE literature [49, 52] found that deeper layers tend to encode coarse-grained global structure while shallower layers capture local details. Our experimental results (Table 4) confirm this pattern, showing that different hierarchical layers contribute complementary information at varying resolutions, improving both expressiveness and controllability in generation.

## 2.3 Connecting Existing Approaches to Representation-Enhanced Generation

Our theoretical formulation provides a general and flexible framework for representation-enhanced generation by introducing two key components: a customizable weighting schedule $\{\alpha_t\}_{t=1}^T$ and a hierarchical latent structure $\{z_l\}_{l=1}^L$. This framework recovers several existing approaches as special cases, in particular RCG [34] and REPA [61].

When $L = 1$, and $\alpha_t = \delta_{t,1}$, i.e. the representations come in fully at the beginning, we have $A_t = 1$, $\forall t$ as a constant. Thus, the hybrid distribution $\tilde{p}_\theta^{\text{constant}}(x_{t-1}|x_t,z;A_t) = p_\theta(x_{t-1}|x_t,z)$,

---

[3]Equation (7) assumes conditional independence of $z_l$'s given $x_0$, i.e., $q(\mathbf{z}^L|x_0) = \prod_l q_l(z_l|x_0)$, though $p_\theta$ must still account for their interactions when inferring from noisy $x_t$. For dependent $z_l$'s given $x_0$, a similar equation applies by replacing $q(z_l|x_0)$ with an appropriate factorization of $q(\mathbf{z}^L|x_0)$ that reflects the specific dependency structure.

and the second log-normalization term in Equation (4) equal to zero. In this case, generation follows a two-stage procedure: first, a latent representation $z$ is sampled from the generative model $p_\theta(z)$, approximating the true marginal $q(z)$; second, the sequence $x_{T-1}, ..., x_0$ is generated conditionally on the sampled $\hat{z} \sim p_\theta(z)$. This formulation corresponds to the two-stage design in RCG [34], where representation modeling and data generation are decoupled. However, such decoupling poses challenges for learning $p_\theta(z)$, particularly when $z$ is high-dimensional or encodes rich semantic information. Since $z$ must be inferred independently of $x_t$, the model is unable to leverage context from the current diffusion state during inference, limiting its expressiveness and sample efficiency.

In contrast, when $L = 1$ and $\alpha_t = 1/T$, $\forall\ t$, the cumulative weights decrease linearly over time, yielding $A_t = 1 - t/T$. In this setting, the hybrid distribution $\tilde{p}_\theta^{\text{linear}}(x_{t-1}|x_t, z; A_t) \propto p_\theta(x_{t-1}|x_t, z)^{1-t/T} p_\theta(x_{t-1}|x_t)^{t/T}$, reflecting that the influence of the representation on $x$-generation gradually increases as the noise level decreases (i.e., as $t$ becomes smaller). However, this setup leads to a training-inference mismatch. During training, the conditional model $p_\theta(x_{t-1}|x_t, z)$ receives the ground truth representation $z \sim q(z|x_0)$ as expectations in Equation (4) are taken over the data distribution $q(\cdot)$, while during inference this representation is not available. Instead, the model must rely on an estimate based on the current available $x_t$, i.e., $\hat{\mu}_t^z$ as the mean of $p_\theta(z|x_t)$. Notably, this estimation becomes more accurate at smaller $t$, when $x_t$ contains less noise, which naturally matches the increased reliance on $z$ in the hybrid distribution $\tilde{p}_\theta^{\text{linear}}$ as $t$ decreases.

To address this mismatch, we approximate the hybrid distribution $\tilde{p}_\theta^{\text{linear}}(x_{t-1}|x_t, z; A_t)$ using $p_\theta(x_{t-1}|x_t, \hat{\mu}_t^z)$ throughout both training and inference. This allows the model to progressively infer $z$ from $x_t$ through $p_\theta(z|x_t)$, ensuring that the representation guidance becomes more prominent as the process denoises, in line with the linear schedule (see Appendix C for details). The second log-normalization term in the objective, which empirically is much smaller than the main denoising loss as supported by our experiments (Appendix F.1), can be omitted from training–it also suffers from training-inference discrepancies. Further, under the von Mises–Fisher distribution assumption, the KL divergence in the third term simplifies to the cosine similarity between mean vectors of the respective distributions (see Appendix C for details). Altogether, this results in the REPA framework [61], which unifies representation extraction and data generation within a single coherent architecture via a representation alignment loss, and different parts of the network specialize at different stages of the generation process. Our experiments (Appendix F.1) validate the above-mentioned approximations, showing that training with the exact $\mathcal{L}_{\text{VB}}^z$ (Equation (3)) achieves similar performance to REPA, while the latter offers a concise objective, simpler architectures, and lower computational cost.

Our method in Section 3 builds upon the REPA setting, adopting the linear weighting schedule and ensuring end-to-end coherence between training and inference. Importantly, we move beyond the original $L = 1$ setting to the more general case of $L > 1$, enabling the novel use of richer multimodal representations through synthetic data. Additionally, inspired by our theoretical framework, we propose a better training curriculum that dynamically balances the data generation and representation modeling objectives, further enhancing model performance and flexibility.

# 3    REED: Representation-Enhanced Elucidation of Diffusion

Building on the insights provided by our generic analytic framework, we present two novel strategies: integrating multimodal representations with synthetic data, and designing an improved training curriculum. We further showcase the versatility of our approach by introducing three practical instantiations across diverse domains, spanning image, protein, and molecule generation.

## 3.1    Multimodal Representations with Synthetic Data

As discussed in Section 2.2, the effectiveness of representation-enhanced generation depends on both the quality of pretrained representations and the interplay between hierarchical levels $\{z_l\}_{l=1}^L$. To effectively harness the information from different sources, we propose integrating representation $z^y$ in a secondary modality $y$ alongside the representation $z^x$ of the primary modality $x$. Representations $z^y$ that are both helpful for score estimation and complementary to $z^x$ yield the greatest improvement.

This cross-modal approach enhances generation quality by leveraging distinct yet related information encoded in different perceptual channels. It also allows integration of different embedding spaces (e.g., VAE space for latent diffusion and DINO embedding space for images), leading to an optimal

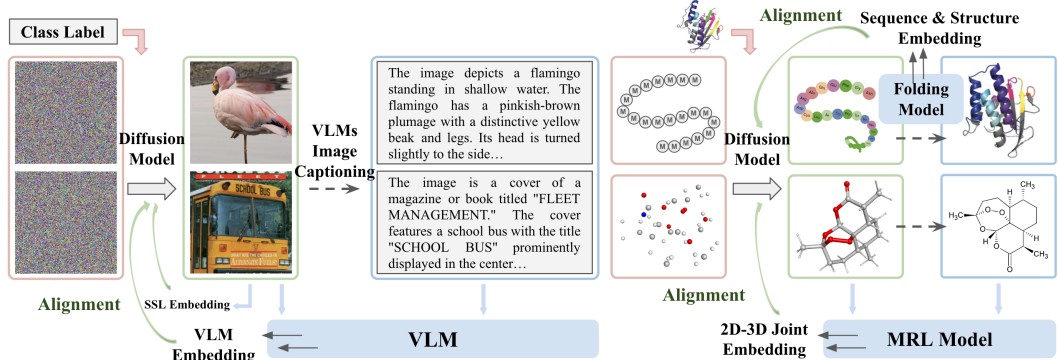

Figure 2: We use synthetic auxiliary data modalities and multimodal representations to enhance representation alignment in diffusion model training. Shown are examples from class-conditional image generation (*left*), protein inverse folding (*top tight*), and molecule generation (*bottom right*).

shared solution space. Such advantage also aligns with the Platonic representation hypothesis [23], which suggests that representations learned on different data and modalities tend to converge toward a unified model of reality. Consequently, with finite available data, incorporating data from other modalities should enhance generation or comprehension in the original modality of interest.

Utilizing the representations $z^x(x)$ and $z^y(y)$ of the data pair $(x, y)$, the resulting representation generation loss, corresponding to the third term in $\mathcal{L}_{\text{VB}}^{\mathbf{z}^L}$, is:

$$\mathcal{L}_{\text{repgen}}(\theta, \psi_x, \psi_y) := -\mathbb{E}_{x,y,t,\epsilon}\left[\lambda_x \text{sim}(z^x(x), \text{proj}^x_{\psi_x}(h_t^{l_x})) + \lambda_y \text{sim}(z^y(y), \text{proj}^y_{\psi_y}(h_t^{l_y}))\right] \quad (8)$$

where $h_t^{l_x}$ and $h_t^{l_y}$ represent the features extracted from the $l_x$-th and $l_y$-th layers of the diffusion transformer encoder $h_t := f_\theta(x_t)$ ($x_t$ denotes the noisy input[4] to the diffusion model at time step $t$), $\text{proj}^x_{\psi_x}$ and $\text{proj}^y_{\psi_y}$ are trainable projection heads parameterized by $\psi_x$ and $\psi_y$, $\text{sim}(\cdot, \cdot)$ is a predefined similarity function (we use cosine similarity in practice), and hyperparameters $\lambda_x, \lambda_y > 0$ controls the relative importance of aligning each representation.

However, directly using multimodal data requires paired samples across modalities, which are often scarce and costly to obtain. In the context of conditional generation, where the model generates outputs based on a specific conditioning variable (e.g., class label or input sequence), it may also introduce shortcuts between the conditioning variable and the representations. For instance, aligning the representation of the input condition as an additional modality may cause the model to simply replicate the condition, rather than extract meaningful features. To address this issue, we generate synthetic multimodal data $y := f_\phi(x)$ using an auxiliary multimodal model $f_\phi$ that bridges modalities $x$ and $y$ (see Figure 2). This introduces stochasticity and diversity into the data pairs, enabling efficient data augmentation that fosters model generalization and mitigates shortcut risks.

In practice, we use multimodal models (between $x$ and $y$) instead of single-modality models to obtain the representations $z^y$ for other auxiliary modalities because their representations are inherently aligned across modalities $x$ and $y$. For images, we use text as an auxiliary modality, generate captions with Vision Language Models (VLMs), and extract multimodal features from intermediate layers of the VLMs. For protein inverse folding, we generate auxiliary structures with folding models and use their single, pair, and structure embeddings as representations. For 3D molecules, we treat 2D graph structure as an auxiliary modality, using external tookits like RDKit to extract atom and bond features. We precompute and store these representations before diffusion training, so REED incurs minimal additional computational overhead compared to REPA. Further analysis of computational cost is provided in Appendix E.1.

## 3.2 Curriculum of Representation Learning and Diffusion Training

We analyze optimal curricula between the two components, representation modeling and data generation, as introduced in Section 2.1. We contend that in REPA [61] type of representation-aligned

---

[4]Noisy VAE features for images, or directly noisy data for proteins and molecules.

diffusion model training, integrating pretrained representations is particularly useful in the early stage of diffusion training. This is also supported by our experimental observation that decaying or dropping REPA loss in the late plateau of training phase yields comparable or even superior performance. Our theoretical analysis in Section 2 provides key insights into the phenomenon. Recall the training loss bound in Equation (4), where the first term is for the approximate conditional distribution based on estimated representations $\hat{\mu}_t^z$, and the last term forces the model to predict ground truth representations. Intuitively, better representation estimations $\hat{\mu}_t^z$ naturally benefit representation-conditioned distribution modeling in the first term.

To leverage these insights, we introduce a joint training curriculum for $\mathcal{L}_{\mathrm{VLB}}^{z^L}$ with special emphasis on representation generation in the initial training phase–rather than adopting a rigid two-stage approach which may suffer from disjoint optimization dynamics and misaligned representation spaces (see more discussions in Appendix D). Specifically, the resulting REED loss at epoch $n$ is given by:

$$\mathcal{L}_{\mathrm{REED}}^n = \alpha(n) \cdot \mathcal{L}_{\mathrm{diffusion}} + \beta(n) \cdot \mathcal{L}_{\mathrm{repgen}} \tag{9}$$

Here, the diffusion loss weight $\alpha(n)$ progressively increases from zero following a phase-in protocol, while the representation alignment weight $\beta(n)$ remains fixed or decays. This curriculum enables a coherent training flow. Moreover, by exposing the model to imperfect $\hat{\mu}_t^z$ early on, we encourage it to learn under noisy representation conditions, which can improve its generalization and robustness. The reasonability of our arguments and the effectiveness of the proposed schedule are well supported by experimental evidence.

### 3.3 Instantiations of the Framework for Domain Representations

In this section, we consider several application areas and incorporate additional inductive biases into the design of representation learning techniques. Our designs demonstrate the effectiveness of representation learning methods for Euclidean, sequential, and graph-structured/3D point-cloud data. To the best of our knowledge, we are the first to apply representation alignment to diffusion models in the context of protein inverse folding and molecule generation.

**Images.**  As discussed in Section 3.1, we adopt multimodal representations–combining image and text embeddings–to enhance single-modal diffusion model training. Image features are drawn from pretrained self-supervised models to offer low-level details, while VLMs generate synthetic captions and corresponding cross-modal embeddings for high-level semantic guidance. In particular, for better alignment between different representation spaces, we use pooled image and text embeddings from VLMs, rather than relying solely on pure text embeddings in single-modal LLMs.

Our analysis in Section 2.2 suggests that shallow latent layers capture fine details, whereas deeper layers represent global semantics. Accordingly, we align shallow layers in the diffusion transformer with SSL image patch embeddings, and deeper layers with pooled VLM embeddings. This aligns with empirical observations: due to the discrepancy between the latent space of external models and the diffusion output (VAE) space, local regularization should be performed in shallow layers as observed in [61]; however, global alignment via pooled embeddings is relatively light and can be applied to deeper layers for improved semantic control.

**Proteins.**  We focus on protein sequence design conditioned on the protein 3D backbone structure – protein inverse folding – which can be framed as a conditional generation task using discrete diffusion models. To expand structural variability and avoid potential shortcuts mentioned in Section 3.1, we employ a folding model, AlphaFold3 (AF3) [1], to generate auxiliary structures from target sequences. Since diffusion sampling with AF3 introduces detailed variability into synthetic structures, using the structure representation in the AF3 diffusion head can improve generalization and controllability of our inverse folding model. We obtain sequence representations from both single token embeddings and pair embeddings from AF3's Pairformer, where the latter models residue interactions.

Since protein inverse folding can be regarded as the reverse of the folding process, we align layers of our discrete diffusion model with those of folding models in the reverse order. Specifically, early encoder layers in our discrete diffusion model align with representations from folding decoders (guided by structure), while deeper decoder layers of our denoising network are trained to match single and pair sequence encoder representations from folding models.

**Molecules.**  Generative models for molecules must respect key symmetries, such as permutation symmetry ($S(N)$) for 2D graphs and Euclidean symmetries ($E(3)$ or $SE(3)$) for 3D structures.

Table 1: FID comparisons with vanilla SiTs and REPA of class-conditional generation on ImageNet $256 \times 256$. No classifier-free guidance (CFG) is used. Iter. denotes training iteration.

| Model | #Params | Iter. | FID↓ |
|---|---|---|---|
| SiT-B/2 | 130M | 400K | 33.0 |
| + REPA | 130M | 400K | 24.4 |
| + REED (ours) | 130M | 400K | **19.5** |
| SiT-L/2 | 458M | 400K | 18.8 |
| + REPA | 458M | 400K | 9.7 |
| + REED (ours) | 458M | 400K | **8.9** |
| SiT-XL/2 | 675M | 400K | 17.2 |
| + REPA | 675M | 150K | 13.6 |
| + REED (ours) | 675M | 150K | **11.6** |
| SiT-XL/2 | 675M | 7M | 8.3 |
| + REPA | 675M | 400K | 7.9 |
| + REPA | 675M | 1M | 6.4 |
| + REPA | 675M | 4M | 5.9 |
| + REED (ours) | 675M | 300K | 8.2 |
| + REED (ours) | 675M | 400K | 7.3 |
| + REED (ours) | 675M | 1M | **4.7** |

Table 2: System-level comparison on ImageNet $256 \times 256$ with CFG. Models with additional CFG scheduling [30] are marked with an asterisk (*).

| Model | Epochs | FID↓ | IS↑ | Pre.↑ | Rec.↑ |
|---|---|---|---|---|---|
| *Pixel diffusion* | | | | | |
| ADM-U | 400 | 3.94 | 186.7 | 0.82 | 0.52 |
| VDM++ | 560 | 2.40 | 225.3 | - | - |
| Simple diffusion | 800 | 2.77 | 211.8 | - | - |
| CDM | 2160 | 4.88 | 158.7 | - | - |
| *Latent diffusion, U-Net* | | | | | |
| LDM-4 | 200 | 3.60 | 247.7 | 0.87 | 0.48 |
| *Latent diffusion, Transformer + U-Net hybrid* | | | | | |
| U-ViT-H/2 | 240 | 2.29 | 263.9 | 0.82 | 0.57 |
| DiffiT* | - | 1.73 | 276.5 | 0.80 | 0.62 |
| MDTv2-XL/2* | 1080 | 1.58 | 314.7 | 0.79 | 0.65 |
| *Latent diffusion, Transformer* | | | | | |
| MaskDiT | 1600 | 2.28 | 276.6 | 0.80 | 0.61 |
| SD-DiT | 480 | 3.23 | - | - | - |
| DiT-XL/2 | 1400 | 2.27 | 278.2 | **0.83** | 0.57 |
| SiT-XL/2 | 1400 | 2.06 | 270.3 | 0.82 | 0.59 |
| + REPA | 200 | 1.96 | 264.0 | 0.82 | 0.60 |
| + REPA | 800 | 1.80 | **284.0** | 0.81 | 0.61 |
| + REED (ours) | **200** | **1.80** | 267.5 | 0.81 | **0.61** |

Following [35], we use graph-level invariant representations from pretrained molecular representation learning (MRL) model to improve generation quality with minimal computation overhead. We adopt diffusion and flow matching models with equivariant GNN backbones to maintain these symmetries, and incorporate lightweight graph-level invariant representations from pretrained $SE(3)$-invariant molecular encoders. Instead of the strictly two-stage representation-conditioned generation in [35], we show that directly aligning diffusion models with pretrained global representations significantly accelerates training and enhances generation quality.

## 4 Experiments

We evaluate the effectiveness of our proposed REED through extensive experiments. The superior performance across image, protein and molecule domains validates the universal benefit of our framework for both continuous and discrete diffusion or flow-matching models.

### 4.1 Image Generation

**Experimental Setup.** For continuous diffusion on Euclidean data, we test REED on class-conditional image generation following the setup in REPA [61]. We conduct experiments on ImageNet [12] at $256 \times 256$ resolution, adopting ADM [13] data preprocessing and encoding each image into a latent vector with the Stable Diffusion VAE [46]. We utilize the B/2, L/2, and XL/2 architectures from SiT [40]. For sampling, consistent with SiT and REPA, we employ the SDE Euler-Maruyama sampler and fix the diffusion timesteps to 250 for all runs. We report Fréchet inception distance (FID [18]), inception score (IS [47]), precision (Pre.) and recall (Rec.) [29] using 50,000 samples for evaluation.

We use Qwen2-VL [55] 2B to generate a synthetic caption for each image and compute joint image-text representations with Qwen2-VL 7B, using both the image and caption as input. We extract the 15th-layer representations from Qwen2-VL 7B, averaging across all tokens for a high-level joint embedding, and align this with the averaged SiT image latents using a cosine similarity loss and MLP projector. We choose the 15th layer for its strong semantic representation; an ablation of different layers is provided in Appendix F.1. For training curriculum (Equation (9)), we keep $\beta(n)$ constant and linearly increase $\alpha(n)$ over the first 50K training iterations, after which it remains fixed for the rest of training. Detailed experimental settings are provided in E.1.

**Baselines.** We compare against recent diffusion-based generation baselines with varying inputs and architectures (details in Appendix E.1): (a) pixel diffusion models (ADM [13], VDM++ [27], Simple diffusion [22], CDM [21]), (b) latent diffusion with U-Net (LDM [46]), (c) latent diffusion using

transformer+U-Net hybrids (U-ViT-H/2 [4], DiffiT [16], MDTv2-XL/2 [15]), and (d) latent diffusion with transformers (MaskDiT [63], SD-DiT [65], DiT [43], SiT [40], and SiT with REPA [61]).

**Overall Results.** We provide FID comparisons in Table 1. REED consistently outperforms vanilla SiT and REPA at the same training iterations across all SiT model scales. In particular, REED achieves FID=8.2 on SiT-XL/2 at 300K steps, surpassing vanilla SiT-XL at 7M steps, and reaches FID=4.7 at 1M steps, outperforming REPA at 4M. Furthermore, we provide comparison between REED and other recent diffusion model for images using classifier-free guidance (CFG [19]). As shown in Table 2, REED matches REPA performance with $4\times$ fewer epochs, achieving FID=1.80 at just 200 epochs. Further details and additional metrics are provided in Appendix F.1.

**Ablation Study on Each Algorithm Component.** To examine the effect of each component of REED, including leveraging multimodal representations and applying the curriculum training schedule, we experiment on the SiT-B/2 architecture and report the results for models trained for 200K iterations and not use classifier-free guidance for generation. As shown in Table 3, removing each component leads to inferior performance than the full REED model, with FID increasing from 24.6 to 29.4 when removing the curriculum schedule and to 27.7 when removing multimodal representations. This demonstrates the effectiveness of each component in our proposed REED. Further ablations on different types of curriculum schedule and multimodal representation are provided in Appendix F.1.

Table 3: Ablation study on each component of REED. We report results on SiT-B/2 training for 200K iterations. No classifier-free guidance is used.

| Model (SiT-B/2, 200K Iter.) | FID↓ | IS↑ | Prec.↑ | Rec.↑ |
|---|---|---|---|---|
| + REPA | 33.2 | 43.7 | 0.54 | 0.63 |
| + REED | **24.6** | **62.2** | **0.58** | **0.64** |
| + REED w/o Multimodal Representations | 27.7 | 55.5 | 0.56 | 0.64 |
| + REED w/o Curriculum Schedule | 29.4 | 50.8 | 0.56 | 0.64 |

**Alignment Depth for Multimodal Representations.** We investigate the optimal alignment depth for image (i.e., Dep.-I) and VLM (i.e., Dep.-VL) representations in Table 4. Aligning image features at a shallow layer (i.e., 4) and text at a deeper layer (i.e., 8) yields the best results, consistent with our theoretical analysis (Section 2.2) that shallow layers capture fine details while deeper layers encode high-level semantics. In contrast, for VLM-only alignment, the best performance comes from aligning at shallow layer 4, indicating the above effect is specific to interactions between multi-latent representations.

Table 4: FID comparison at different alignment depth on SiT-B/2.

| Iter. | Dep.-I | Dep.-VL | FID↓ |
|---|---|---|---|
| 200K | 4 | 2 | 25.9 |
| 200K | 4 | 4 | 25.9 |
| 200K | 4 | 6 | 24.8 |
| 200K | 4 | 8 | **24.6** |
| 200K | 4 | 10 | 26.4 |
| 200K | - | 4 | **45.5** |
| 200K | - | 8 | 45.6 |

### 4.2 Protein Sequence Generation

**Experimental Setup.** For discrete diffusion on sequence data, we experiment on protein inverse folding–generating protein sequences conditioned on backbone 3D structure. Following [54], we train an inverse folding model using the Multiflow [7] objective and the ProteinMPNN [11] architecture, with the PDB training set from [11]. We filter for single-chain proteins with sequences under 256 residues, yielding 13,753 training and 811 test proteins.

We use AlphaFold3 [1] to generate auxiliary structures from target sequences, extracting latents from its diffusion head as structure representations, as well as both single and pair embeddings of Pairformer as amino acid representations. We align the ProteinMPNN encoder output with the structure representation, and the decoder's first-layer output with the single and pair token representations, reflecting the reverse nature between inverse folding and folding. Regarding training curriculum, $\beta(n)$ is kept constant, while $\alpha(n)$ linearly increases over the first 50 epochs, then decays with a cosine schedule.

**Evaluations.** We assess inverse folding performance using sequence recovery rate (Seq. Recovery), self-consistency RMSD (scRMSD), and pLDDT from ESMFold [36] on generated sequences. For each test structure, we generate a sequence and report the median scRMSD and pLDDT across the test set. We also compute the proportion of cases with scRMSD below 2Å and pLDDT above 80, standard thresholds for successful inverse folding [7, 41]. Further details of the metrics are in Appendix E.2.

Table 5: Model performance on protein inverse folding. REED achieves high sequence recovery rate and pLDDT and low scRMSD across various epochs and significantly accelerates training. We report the mean across 3 random seeds, with standard deviations in parentheses.

| Methods | Epochs | Seq. Recovery(%) ↑ | scRMSD↓ | %(scRMSD<2) ↑ | pLDDT ↑ | %(pLDDT>80)(%) ↑ |
|---------|--------|--------------------|---------|----------------|---------|-------------------|
| ProteinMPNN [11] | 200 | 41.73 (0.08) | 1.801 (0.035) | 54.45 (0.68) | 82.68 (0.25) | 65.66 (1.39) |
| DiscreteDiff | 50 | 38.29 (0.03) | 2.047 (0.018) | 49.11 (0.29) | 81.52 (0.13) | 58.80 (0.21) |
| + REED (ours) | 50 | 41.06 (0.03) | 1.788 (0.033) | 54.37 (1.09) | 82.74 (0.08) | 67.41 (0.38) |
| DiscreteDiff | 100 | 40.74 (0.04) | 1.789 (0.015) | 54.17 (0.61) | 82.80 (0.08) | 67.57 (0.62) |
| + REED (ours) | 100 | 41.91 (0.07) | **1.780 (0.017)** | **56.23 (0.62)** | 83.09 (0.14) | 68.41 (0.38) |
| DiscreteDiff | 150 | 41.18 (0.08) | 1.828 (0.046) | 54.87 (1.02) | 82.92 (0.08) | 67.25 (1.02) |
| + REED (ours) | 150 | **42.26 (0.12)** | 1.799 (0.016) | 54.66 (0.53) | **83.17 (0.12)** | **69.07 (0.37)** |

**Results.** We compare REED against the de facto inverse folding method ProteinMPNN [11] and and discrete diffusion models without REED (DiscreteDiff), with results in Table 5. REED consistently outperform other methods in all metrics, demonstrating the effectiveness of incorporating powerful representations in the protein domain. Meanwhile, REED accelerates training, reaching a 41.5% sequence recovery rate in just 70 epochs, whereas DiscreteDiff requires 250 epochs. Ablation studies on different combinations among single, pair and structure representations are in Appendix F.2.

### 4.3 Molecule Generation

**Experimental Setup.** For structured and geometric graph data, we choose the 3D molecule generation task with equivariant point-cloud diffusion models. Leveraging symmetry-preserving representations and equivariant backbones, our approach effectively models molecular graph distributions while respecting their inherent symmetries. We adopt the challenging Geom-Drug [3] dataset, which contains large drug-like molecules (average 44 atoms). Following SemlaFlow [25], we filter out molecules with more than 72 atoms in the training set and use standard splits. Our base model is SemlaFlow [25] with an $E(3)$-equivariant backbone and the ODE flow matching framework, and use the Unimol [64] further finetuned on GEOM-DRUG as the pretrained encoder. All other training and evaluation settings follow [25].

**Evaluations.** We adopt standard benchmark evaluation metrics: *atom stability*, *molecule stability*, *validity*, as well as additional metrics *energy*, *strain*, and sampling efficiency (NFE). Full metric details are in Appendix E.3. We compare against state-of-the-art 3D generators that jointly generate atoms and bonds, including MiDi [53], EQGAT-diff [31], and SemlaFlow [25], since those approaches only generating atoms fall short in producing high-quality molecules as is explained in [25]. Notably, we train SemlaFlow+REED for just 200 epochs, compared to 800 epochs for EQGAT-diff.

Table 6: Performance of REED with SemlaFlow as the base model on GEOM-DRUG dataset. Atom stability, molecule stability, and validity are reported as percentages, while energy and strain energy are expressed in kcal·mol$^{-1}$. Results marked with $^*$ were reproduced in our own experiments. All results are calculated based on 5k randomly sampled molecules.

| Methods | Atom Stab ↑ | Mol Stab ↑ | Valid ↑ | Energy ↓ | Strain ↓ | NFE ↓ |
|---------|-------------|------------|---------|----------|----------|-------|
| MiDi | 99.8 | 91.6 | 77.8 | - | - | 500 |
| EQGAT-diff | 99.8 | 93.4 | **94.6** | 148.8 | 140.2 | 500 |
| SemlaFlow$^*$ | 99.88 | 97.4 | 93.4 | 114.44 | 74.18 | **100** |
| + REED (ours) | **99.93** | **98.1** | 94.5 | **105.63** | **65.33** | **100** |

**Results.** As shown in Table 6, SemlaFlow trained with our REED significantly improves almost all metrics compared to the baseline and outperforms other state-of-the-art models. Remarkably, the superior performance is achieved within much less training computation and sampling NFE compared to EQGAT-diff, verifying the effectiveness of REED.

## 5 Conclusion

In this paper, we introduced a general theoretical framework for incorporating representation guidance into diffusion model training. Our proposed design strategies consistently achieve superior performance and accelerated training. across a range of domains. While we adopt a linear representation weighting schedule in our experiments, exploring adaptive schedules remains an intriguing direction for future research. Broader impacts include data efficiency in machine learning and biology.

## Acknowledgement

CW, CZ and TJ acknowledge support from the Machine Learning for Pharmaceutical Discovery and Synthesis (MLPDS) consortium, the NSF Expeditions grant (award 1918839) Understanding the World Through Code, and the DSO Singapore grant on next generation techniques for protein ligand binding. CZ and SB were partly supported by the ARPA-H ADAPT program. SG is supported by the MathWorks Engineering Fellowship. SG and SJ acknowledge the support of the NSF Award CCF-2112665 (TILOS AI Institute), an Alexander von Humboldt fellowship, and Office of Naval Research grant N00014-20-1-2023 (MURI ML-SCOPE).

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

# Appendix

## A   Related Work

**Enhancing Diffusion Models with Pretrained Representations.** Recent empirical studies have explored various strategies for incorporating pretrained representations into diffusion model training. Representation-Conditioned Generation (RCG) [34] addresses unconditional image generation by first training a diffusion model to generate semantic features based on a pretrained self-supervised encoder, and subsequently conditioning a second image diffusion model on these features to achieve high-quality outputs. Representation Alignment (REPA) [61] introduces a regularization technique that aligns intermediate noisy states within the diffusion process to clean image representations extracted from external pretrained visual encoders, thereby improving both representation learning and generation performance. Despite their empirical effectiveness, there remains a gap in the theoretical understanding of how pretrained representations can enhance diffusion model training. Additionally, they focus primarily on image generation, leveraging visual encoder representations, without considering a diverse array of modalities or a wide range of application domains.

**Unifying Representation Learning and Diffusion Models.** Many recent works seek to bridge representation learning and diffusion-based generation. Several works leverage or refine representations obtained from diffusion models. For example, [60] proposes a hybrid framework capable of performing both discriminative tasks and diffusion-based generation, while [58] uncovers the discriminative nature of intermediate representations within diffusion models. Subsequent studies, such as [59, 33], further distill these learned representations for various downstream applications. Additionally, denoising objectives have been harnessed to enhance representation learning; for instance, [10] deconstructs diffusion models to advance denoising-based representation learning. Diffusion models have also benefited from auxiliary self-supervised learning signals–[65] demonstrates that incorporating an auxiliary discriminative self-supervised loss can strengthen diffusion model training. In contrast to these approaches, our focus is on improving diffusion model training by utilizing high-quality pretrained representations from a variety of modalities.

**Generative Models for Different Domains.** Diffusion-based generative models have demonstrated remarkable performance across a diverse range of data types and application areas. In image generation–where data is continuous and resides in Euclidean space–denoising transformers such as DiT [43] (Diffusion Transformer) and SiT [40] (which incorporates continuous stochastic interpolants [2] to enhance DiT) have been widely adopted. For protein sequence generation, which involves discrete sequential data, discrete diffusion and flow models have been successfully applied, as seen in works such as [7, 41, 54]. In the domain of molecular generation, where data is represented as structured and geometric graphs, $E(3)$-equivariant generative approaches have been developed to model 3D molecular structures. Notable examples include the MiDi model [53] for joint 2D and 3D denoising diffusion, EQGAT-diff [31] for 3D diffusion, and SemlaFlow [25], an ODE flow matching model for 3D molecule data. Despite their widespread use, training diffusion-based generative models remains computationally intensive, often requiring significant resources and time. This poses challenges for broader adoption, especially in domains with complex and high-dimensional data.

## B   Preliminaries

Our theoretical analysis in Appendix C.2 builds on the variational framework of denoising diffusion probabilistic models (DDPM) [20, 48]. Below, we briefly summarize the key formulas and concepts; for further details, we refer readers to the original papers.

Diffusion models can be formulated as latent variable models, $p_\theta(x_0) := \int p_\theta(x_{0:T})dx_{1:T}$, where latent variables $x_{1:T}$ are generated by a forward diffusion process $q(x_{1:T}|x_0)$ that forms a Markov chain $x_T - x_{T-1} - ... - x_1 - x_0$, with a predefined variance schedule $\beta_1, ..., \beta_T$:

$$q(x_{1:T}|x_0) := \prod_{t=1}^{T} q(x_t|x_{t-1}), \quad q(x_t|x_{t-1}) := \mathcal{N}(x_t; \sqrt{1-\beta_t}x_{t-1}, \beta_t\mathbf{I}) \tag{10}$$

A variational upper bound for $-\mathbb{E}[\log p_\theta(x_0)]$ is then given by

$$-\mathbb{E}_{q(x_0)} \log p_\theta(x_0) \le \mathbb{E}_{q(x_{0:T})} \log \frac{q(x_{1:T}|x_0)}{p_\theta(x_{0:T})} := \mathcal{L}_{\text{VB}} \tag{11}$$

where $\mathcal{L}_{\text{VB}}$ can be decomposed as

$$\mathcal{L}_{\text{VB}} = \mathbb{E}_{q(x_{0:T})} \left[ \log \frac{q(x_T|x_0) \prod_{t=2}^{T} q(x_{t-1}|x_t, x_0)}{p_\theta(x_T) \prod_{t=1}^{T} p_\theta(x_{t-1}|x_t)} \right] \tag{12}$$

$$= \mathbb{E}_{q(x_{0:T})} \left[ D_{\text{KL}}(q(x_T|x_0)||p_\theta(x_T)) + \sum_{t=2}^{T} D_{\text{KL}}(q(x_{t-1}|x_t, x_0)||p_\theta(x_{t-1}|x_t)) - \log p_\theta(x_0|x_1) \right] \tag{13}$$

The reverse process $p_\theta(x_{t-1}|x_t)$ is typically parameterized as a Gaussian $\mathcal{N}(x_{t-1}; \mu_\theta(x_t, t), \sigma_t^2 \mathbf{I})$ as in [20], enabling tractable computation of the KL divergences.

In Appendix C.2, we extend this framework by introducing an additional latent variable $z$, representing a high-quality pretrained representation of $x_0$. We show how incorporating such semantic representations can enhance diffusion model training.

Note that there are other variants of diffusion models from a different theoretical perspective, such as score matching [50], stochastic interpolants [2], and flow matching [38]. Due to their equivalence with the denoising diffusion models framework, our theoretical insights in Section 2 are broadly applicable to these approaches. In our experiments, we adopt stochastic interpolants (as in SiT [40]) for image generation, discrete flow models (as in Multiflow [7]) for protein sequence generation, and $E(3)$-equivariant flow matching (as in SemlaFlow [25]) for molecule generation.

## C  Additional Theoretical Results

### C.1  Additional Theoretical Characterizations and Proofs for Section 2

#### C.1.1  Derivation of Equation (2)

Similar to the variational upper bound of DDPM, by incorporating additional latent variable $z$ following the Markov chain $x_T - x_{T-1} - ... - x_1 - x_0 - z$, we have

$$-\log p_\theta(x_0) \le -\log p_\theta(x_0) + D_{\text{KL}}(q(x_{1:T}, z|x_0)||p_\theta(x_{1:T}, z|x_0)) \tag{14}$$

$$= -\log p_\theta(x_0) + \mathbb{E}_{q(x_{1:T}, z|x_0)} \left[ \log \frac{q(x_{1:T}, z|x_0)}{p_\theta(x_{0:T}, z)} + \log p_\theta(x_0) \right] \tag{15}$$

$$= \mathbb{E}_{q(x_{1:T}, z|x_0)} \left[ \log \frac{q(x_{1:T}, z|x_0)}{p_\theta(x_{0:T}, z)} \right] \tag{16}$$

Therefore,

$$-\mathbb{E}_{q(x_0)} \log p_\theta(x_0) \le \mathbb{E}_{q(x_{0:T}, z)} \left[ \log \frac{q(x_{1:T}, z|x_0)}{p_\theta(x_{0:T}, z)} \right] := \mathcal{L}_{\text{VB}}^z \tag{17}$$

Further decomposing the numerator of $\mathcal{L}_{\text{VB}}^z$ leads to:

$$\mathcal{L}_{\text{VB}}^z = \mathbb{E}_{q(x_{0:T}, z)} \left[ \log \frac{q(z|x_0)q(x_{1:T}|x_0)}{p_\theta(x_{0:T}, z)} \right] \tag{18}$$

$$= \mathbb{E}_{q(x_{0:T}, z)} \left[ \log \frac{q(z|x_0) \cdot \prod_{t=1}^{T} q(x_{t-1}|x_t, x_0) \cdot q(x_T|x_0)}{p_\theta(x_{0:T}, z)} \right] \tag{19}$$

Note that when $t = 1$, $q(x_{t-1}|x_t, x_0) = q(x_0|x_1, x_0) = 1$.

### C.1.2 Derivation of Equation (3)

The marginal distribution $p(x_{0:T}, z)$ can be decomposed as

$$p(x_{0:T}, z) = p(x_T) \cdot \prod_{i=t+1}^{T} p(x_{i-1}|x_{i:T}) \cdot p(z|x_{t:T}) \cdot \prod_{i=1}^{t} p(x_{i-1}|x_{i:T}, z) \tag{20}$$

following the order $x_T, ..., x_t, z, x_{t-1}, ..., x_0$ for any $t = 0, 1, ..., T$.[5]

Based on the Markov chain $x_T - x_{T-1} - ... - x_1 - x_0 - z$, the following conditional independencies hold: $p(x_{i-1}|x_{i:T}) = p(x_{i-1}|x_i)$, $p(x_{i-1}|x_{i:T}, z) = p(x_{i-1}|x_i, z)$, and $p(z|x_{t:T}) = p(z|x_t)$. Thus, for any $t = 0, 1, ..., T$, we can exactly decompose $p(x_{0:T}, z)$ as follows:

$$p(x_{0:T}, z) = p(z|x_0) \cdot p(x_0|x_1) \cdot p(x_1|x_2) \cdot ... \cdot p(x_{T-1}|x_T) \cdot p(x_T) \tag{21}$$
$$= p(z|x_1) \cdot p(x_0|x_1, z) \cdot p(x_1|x_2) \cdot ... \cdot p(x_{T-1}|x_T) \cdot p(x_T) \tag{22}$$
$$= ...$$
$$= p(z|x_t) \cdot \prod_{i=1}^{t} p(x_{i-1}|x_i, z) \cdot \prod_{i=t+1}^{T} p(x_{i-1}|x_i) \cdot p(x_T) \tag{23}$$
$$= ...$$
$$= p(z|x_{T-1}) \cdot p(x_0|x_1, z) \cdot ... \cdot p(x_{T-2}|x_{T-1}, z) \cdot p(x_{T-1}|x_T) \cdot p(x_T) \tag{24}$$
$$= p(z|x_T) \cdot p(x_0|x_1, z) \cdot p(x_1|x_2, z) \cdot ... \cdot p(x_{T-1}|x_T, z) \cdot p(x_T) \tag{25}$$

### C.1.3 Proof for Proposition 1

**Proposition 1** (Decomposition Structure of the Variational Bound). *Let $\{\alpha_t \geq 0\}_{t=1}^{T}$ be a set of weights summing to one, and define $A_t \in [0, 1] := \sum_{i=t}^{T} \alpha_i$. Then, the variational bound $\mathcal{L}_{VB}^{z}$ in Equation (2) can be written as:*

$$\mathcal{L}_{VB}^{z} = \sum_{t=1}^{T} \mathbb{E}_{q(x_{0:T}, z)} \left[ \log \frac{q(x_{t-1}|x_t, x_0)}{\tilde{p}_\theta(x_{t-1}|x_t, z; A_t)} \right] - \sum_{t=1}^{T} \mathbb{E}_{q(x_{0:T}, z)} \left[ \log Z_t(x_t, z; A_t) \right]$$
$$+ \sum_{t=1}^{T} \alpha_t \mathbb{E}_{q(x_{0:T})} \left[ D_{KL}(q(z|x_0)||p_\theta(z|x_t)) \right] \tag{26}$$

*where $\tilde{p}_\theta(x_{t-1}|x_t, z; A_t)$ and its normalization $Z_t(x_t, z; A_t)$ are defined as:*

$$\tilde{p}_\theta(x_{t-1}|x_t, z; A_t) = \frac{1}{Z(x_t, z)} p_\theta^{A_t}(x_{t-1}|x_t, z) p_\theta^{1-A_t}(x_{t-1}|x_t) \tag{27}$$

$$Z_t(x_t, z; A_t) = \int p_\theta^{A_t}(x_{t-1}|x_t, z) p_\theta^{1-A_t}(x_{t-1}|x_t) dx_{t-1} \tag{28}$$

*Proof.* The marginal distribution $p_\theta(x_{0:T}, z)$, which appears in the denominator of $\mathcal{L}_{VB}^{z}$ (Equation (2)), can be decomposed as shown in Equation (3) for any $t = 1, ..., T$. By aggregating these decompositions according to the weight schedule $\{\alpha_t\}_{t=1}^{T}$, we obtain an equivalent form of $\mathcal{L}_{VB}^{z}$:

---

[5]When $t = 0$ or $t = T$, the corresponding products in Equation (20) reduce to 1.

$$\mathcal{L}_{\text{VB}}^z = \sum_{t=1}^T \alpha_t \mathbb{E}_{q(x_{0:T},z)} \left[ \log \frac{q(z|x_0) \cdot \prod_{t=1}^T q(x_{t-1}|x_t,x_0) \cdot q(x_T|x_0)}{p(z|x_t) \cdot \prod_{i=1}^t p(x_{i-1}|x_i,z) \cdot \prod_{i=t+1}^T p(x_{i-1}|x_i) \cdot p(x_T)} \right] \tag{29}$$

$$= \sum_{t=1}^T \alpha_t \mathbb{E}_{q(x_{0:T},z)} \left[ \log \frac{q(z|x_0)}{p_\theta(z|x_t)} + \sum_{i=1}^t \log \frac{q(x_{i-1}|x_i,x_0)}{p_\theta(x_{i-1}|x_i,z)} \right. \tag{30}$$

$$\left. + \sum_{i=t+1}^T \log \frac{q(x_{i-1}|x_i,x_0)}{p_\theta(x_{i-1}|x_i)} + \log \frac{q(x_T|x_0)}{p_\theta(x_T)} \right] \tag{31}$$

$$= \sum_{t=1}^T \mathbb{E}_{q(x_{0:T},z)} \left[ \left( \sum_{i=t}^T \alpha_i \right) \log \frac{q(x_{t-1}|x_t,x_0)}{p_\theta(x_{t-1}|x_t,z)} + \left( \sum_{i=1}^{t-1} \alpha_i \right) \log \frac{q(x_{t-1}|x_t,x_0)}{p_\theta(x_{t-1}|x_t)} \right] \tag{32}$$

$$+ \sum_{t=1}^T \alpha_t \mathbb{E}_{q(x_{0:T},z)} \left[ \log \frac{q(z|x_0)}{p_\theta(z|x_t)} \right] + \mathbb{E}_{q(x_{0:T},z)} \left[ \log \frac{q(x_T|x_0)}{p_\theta(x_T)} \right] \tag{33}$$

$$= \sum_{t=1}^T \mathbb{E}_{q(x_{0:T},z)} \left[ A_t \log \frac{q(x_{t-1}|x_t,x_0)}{p_\theta(x_{t-1}|x_t,z)} + (1 - A_t) \log \frac{q(x_{t-1}|x_t,x_0)}{p_\theta(x_{t-1}|x_t)} \right] \tag{34}$$

$$+ \sum_{t=1}^T \alpha_t \mathbb{E}_{q(x_{0:T},z)} \left[ \log \frac{q(z|x_0)}{p_\theta(z|x_t)} \right] + \mathbb{E}_{q(x_{0:T},z)} \left[ \log \frac{q(x_T|x_0)}{p_\theta(x_T)} \right] \tag{35}$$

$$= \sum_{t=1}^T \mathbb{E}_{q(x_{0:T},z)} \left[ \log \frac{q(x_{t-1}|x_t,x_0)}{p_\theta^{A_t}(x_{t-1}|x_t,z) p_\theta^{1-A_t}(x_{t-1}|x_t)} \right] \tag{36}$$

$$+ \sum_{t=1}^T \alpha_t \mathbb{E}_{q(x_{0:T},z)} \left[ \log \frac{q(z|x_0)}{p_\theta(z|x_t)} \right] + \mathbb{E}_{q(x_{0:T},z)} \left[ \log \frac{q(x_T|x_0)}{p_\theta(x_T)} \right] \tag{37}$$

Since $p_\theta(x_T)$ is a pre-specified fixed distribution (typically standard Gaussian) that is easy to sample from, and $q(x_T|x_0)$ converges to this Gaussian as $T$ is large, the last term is approximately zero. Additionally, it is not involved in the optimization of model parameters $\theta$, thus it can be omitted from $\mathcal{L}_{\text{VB}}^z$. Following the definitions of $\tilde{p}_\theta(x_{t-1}|x_t,z;A_t)$ and $Z_t(x_t,z;A_t)$, the variational bound $\mathcal{L}_{\text{VB}}^z$ can then be written as:

$$\mathcal{L}_{\text{VB}}^z = \sum_{t=1}^T \mathbb{E}_{q(x_{0:T},z)} \left[ \log \frac{q(x_{t-1}|x_t,x_0)}{\frac{1}{Z_t(x_t,z;A_t)} p_\theta^{A_t}(x_{t-1}|x_t,z) p_\theta^{1-A_t}(x_{t-1}|x_t)} \right] \tag{38}$$

$$- \sum_{t=1}^T \mathbb{E}_{q(x_{0:T},z)} \left[ \log Z_t(x_t,z;A_t) \right] + \mathbb{E}_{q(x_{0:T})} \mathbb{E}_{q(z|x_0)} \left[ \log \frac{q(z|x_0)}{p_\theta(z|x_t)} \right] \tag{39}$$

$$= \sum_{t=1}^T \mathbb{E}_{q(x_{0:T},z)} \left[ \log \frac{q(x_{t-1}|x_t,x_0)}{\tilde{p}_\theta(x_{t-1}|x_t,z;A_t)} \right] - \sum_{t=1}^T \mathbb{E}_{q(x_{0:T},z)} \left[ \log Z_t(x_t,z;A_t) \right]$$

$$+ \sum_{t=1}^T \alpha_t \mathbb{E}_{q(x_{0:T})} \left[ D_{\text{KL}}(q(z|x_0)||p_\theta(z|x_t)) \right] \tag{40}$$

$\square$

### C.1.4  Bounds on the Log-Normalization Term

To clarify the role of the second log-normalization term in the expression for $\mathcal{L}_{\text{VB}}^z$ (Equation (4)), we present the following proposition, which provides both a bound and a closed-form solution under the Gaussian assumption for the reverse process:

**Proposition 2** (Bounds on the Log-Normalization Term). *The log-normalization term in Equation* (4) *satisfies:*

$$0 \leq -\sum_{t=1}^{T} \mathbb{E}_{q(x_{0:T}, z)} \Big[ \log Z_t(x_t, z; A_t) \Big] \leq \min \Bigg\{ \sum_{t=1}^{T} \mathbb{E}_{q(x_{0:T}, z)} A_t D_{KL}(p_\theta(x_{t-1}|x_t) || p_\theta(x_{t-1}|x_t, z)),$$

$$\sum_{t=1}^{T} \mathbb{E}_{q(x_{0:T}, z)} (1 - A_t) D_{KL}(p_\theta(x_{t-1}|x_t, z) || p_\theta(x_{t-1}|x_t)) \Bigg\}$$

*Specifically, under the common Gaussian assumption for the reverse process [20], i.e., $p_\theta(x_{t-1}|x_t) \sim \mathcal{N}(\mu_\theta^u(x_t, t), \sigma_t^2)$ and $p_\theta(x_{t-1}|x_t, z) \sim \mathcal{N}(\mu_\theta^c(x_t, z, t), \sigma_t^2)$, the term has the following closed-form expression:*

$$-\sum_{t=1}^{T} \mathbb{E}_{q(x_{0:T}, z)} \Big[ \log Z_t(x_t, z; A_t) \Big] = \sum_{t=1}^{T} \frac{A_t(1 - A_t)}{2\sigma_t^2} \mathbb{E}_{q(x_{0:T}, z)} |\mu_\theta^c(x_t, z, t) - \mu_\theta^u(x_t, t)|^2 \quad (41)$$

*Proof.* First, we consider the lower bound of the log-normalization term. Utilizing Hölder's inequality with exponents $1/A_t$ and $1/(1 - A_t)$, we have

$$Z_t(x_t, z; A_t) = \int p_\theta^{A_t}(x_{t-1}|x_t, z) p_\theta^{1-A_t}(x_{t-1}|x_t) dx_{t-1} \quad (42)$$

$$\leq \left( \int p_\theta(x_{t-1}|x_t, z) dx_{t-1} \right)^{A_t} \left( \int p_\theta(x_{t-1}|x_t) dx_{t-1} \right)^{1-A_t} = 1^{A_t} \cdot 1^{1-A_t} = 1 \quad (43)$$

Thus, this establishes the lower bound:

$$-\sum_{t=1}^{T} \mathbb{E}_{q(x_{0:T}, z)} \Big[ \log Z_t(x_t, z; A_t) \Big] \geq -\sum_{t=1}^{T} \mathbb{E}_{q(x_{0:T}, z)} \Big[ \log 1 \Big] = 0 \quad (44)$$

Second, we investigate its upper bound. Note that $Z_t(x_t, z; A_t)$ can be written as:

$$Z_t(x_t, z; A_t) = \int p_\theta(x_{t-1}|x_t) \left( \frac{p_\theta(x_{t-1}|x_t, z)}{p_\theta(x_{t-1}|x_t)} \right)^{A_t} dx_{t-1} = \mathbb{E}_{p_\theta(x_{t-1}|x_t)} \left[ \left( \frac{p_\theta(x_{t-1}|x_t, z)}{p_\theta(x_{t-1}|x_t)} \right)^{A_t} \right] \quad (45)$$

Leveraging the concavity of the logarithm and Jensen's inequality, we have

$$\log Z_t(x_t, z; A_t) \geq \mathbb{E}_{p_\theta(x_{t-1}|x_t)} \left[ \log \left( \frac{p_\theta(x_{t-1}|x_t, z)}{p_\theta(x_{t-1}|x_t)} \right)^{A_t} \right] \quad (46)$$

$$= A_t \mathbb{E}_{p_\theta(x_{t-1}|x_t)} \left[ \log \frac{p_\theta(x_{t-1}|x_t, z)}{p_\theta(x_{t-1}|x_t)} \right] = -A_t D_{\text{KL}}(p_\theta(x_{t-1}|x_t) || p_\theta(x_{t-1}|x_t, z)) \quad (47)$$

Therefore,

$$-\sum_{t=1}^{T} \mathbb{E}_{q(x_{0:T}, z)} \Big[ \log Z_t(x_t, z; A_t) \Big] \leq \sum_{t=1}^{T} \mathbb{E}_{q(x_{0:T}, z)} A_t D_{\text{KL}}(p_\theta(x_{t-1}|x_t) || p_\theta(x_{t-1}|x_t, z)) \quad (48)$$

Similarly, using the alternative expression of $Z_t(x_t, z; A_t)$ as follows:

$$Z_t(x_t, z; A_t) = \int p_\theta(x_{t-1}|x_t, z) \left( \frac{p_\theta(x_{t-1}|x_t)}{p_\theta(x_{t-1}|x_t, z)} \right)^{1-A_t} dx_{t-1} \quad (49)$$

$$= \mathbb{E}_{p_\theta(x_{t-1}|x_t, z)} \left[ \left( \frac{p_\theta(x_{t-1}|x_t)}{p_\theta(x_{t-1}|x_t, z)} \right)^{1-A_t} \right] \quad (50)$$

we have

$$\log Z_t(x_t, z; A_t) \geq \mathbb{E}_{p_\theta(x_{t-1}|x_t, z)} \left[ \log \left( \frac{p_\theta(x_{t-1}|x_t)}{p_\theta(x_{t-1}|x_t, z)} \right)^{1-A_t} \right] \tag{51}$$

$$= (1 - A_t)\mathbb{E}_{p_\theta(x_{t-1}|x_t, z)} \left[ \log \frac{p_\theta(x_{t-1}|x_t)}{p_\theta(x_{t-1}|x_t, z)} \right] = -(1 - A_t)D_{\text{KL}}(p_\theta(x_{t-1}|x_t, z)||p_\theta(x_{t-1}|x_t)) \tag{52}$$

and subsequently

$$-\sum_{t=1}^{T} \mathbb{E}_{q(x_{0:T}, z)} \Big[ \log Z_t(x_t, z; A_t) \Big] \leq \sum_{t=1}^{T} \mathbb{E}_{q(x_{0:T}, z)}(1 - A_t)D_{\text{KL}}(p_\theta(x_{t-1}|x_t, z)||p_\theta(x_{t-1}|x_t)) \tag{53}$$

Putting both together, we derive the upper bound of the log-normalization term as

$$\min \left\{ \sum_{t=1}^{T} \mathbb{E}_{q(x_{0:T}, z)} A_t D_{\text{KL}}(p_\theta(x_{t-1}|x_t)||p_\theta(x_{t-1}|x_t, z)), \tag{54} \right.$$

$$\left. \sum_{t=1}^{T} \mathbb{E}_{q(x_{0:T}, z)}(1 - A_t)D_{\text{KL}}(p_\theta(x_{t-1}|x_t, z)||p_\theta(x_{t-1}|x_t)) \right\} \tag{55}$$

Finally, we calculate the closed-form expression under the Gaussian assumption for the reverse process $p_\theta(x_{t-1}|x_t)$ and $p_\theta(x_{t-1}|x_t, z)$ and denoting $d$ as the dimensionality of $x_t$, i.e.,

$$p_\theta(x_{t-1}|x_t, z) = \frac{1}{(2\pi\sigma_t^2)^{d/2}} \exp \left( -\frac{|x_{t-1} - \mu_\theta^c(x_t, z, t)|^2}{2\sigma_t^2} \right) \tag{56}$$

$$p_\theta(x_{t-1}|x_t) = \frac{1}{(2\pi\sigma_t^2)^{d/2}} \exp \left( -\frac{|x_{t-1} - \mu_\theta^u(x_t, t)|^2}{2\sigma_t^2} \right) \tag{57}$$

Multiplying them together yields:

$$p_\theta^{A_t}(x_{t-1}|x_t, z)p_\theta^{1-A_t}(x_{t-1}|x_t) = \frac{1}{(2\pi\sigma_t^2)^{d/2}} \exp \left( -\frac{f_\theta(x_{t-1}; x_t, z, A_t)}{2\sigma_t^2} \right) \tag{58}$$

where $f_\theta(x_{t-1}; x_t, z, A_t) := A_t|x_{t-1} - \mu_\theta^c(x_t, z, t)|^2 + (1 - A_t)|x_{t-1} - \mu_\theta^u(x_t, t)|^2$ is the exponent term. For simplicity of notation, use $\mu_c$ to represent $\mu_\theta^c(x_t, z, t)$ and $\mu_u$ to represent $\mu_\theta^u(x_t, t)$. The exponent term $f_\theta(x_{t-1}; x_t, z, A_t)$ can be expanded as:

$$f_\theta(x_{t-1}; x_t, z, A_t) = A_t|x_{t-1} - \mu_c|^2 + (1 - A_t)|x_{t-1} - \mu_u|^2 \tag{59}$$

$$= A_t(x_{t-1} - \mu_c)^T(x_{t-1} - \mu_c) + (1 - A_t)(x_{t-1} - \mu_u)^T(x_{t-1} - \mu_u) \tag{60}$$

$$= A_t(x_{t-1}^T x_{t-1} - 2\mu_c^T x_{t-1} + \mu_c^T \mu_c) + (1 - A_t)(x_{t-1}^T x_{t-1} - 2\mu_u^T x_{t-1} + \mu_u^T \mu_u) \tag{61}$$

$$= x_{t-1}^T x_{t-1} - 2(A_t\mu_c + (1 - A_t)\mu_u)^T x_{t-1} + A_t\mu_c^T \mu_c + (1 - A_t)\mu_u^T \mu_u \tag{62}$$

$$= |x_{t-1} - \mu_{\text{weighted}}|^2 + A_t\mu_c^T \mu_c + (1 - A_t)\mu_u^T \mu_u - \mu_{\text{weighted}}^T \mu_{\text{weighted}} \tag{63}$$

where $\mu_{\text{weighted}} = A_t\mu_c + (1 - A_t)\mu_u$. Denote

$$C := A_t\mu_c^T \mu_c + (1 - A_t)\mu_u^T \mu_u - \mu_{\text{weighted}}^T \mu_{\text{weighted}} \tag{64}$$

Then,

$$p_\theta^{A_t}(x_{t-1}|x_t, z)p_\theta^{1-A_t}(x_{t-1}|x_t) = \frac{1}{(2\pi\sigma_t^2)^{d/2}} \exp \left( -\frac{|x_{t-1} - \mu_{\text{weighted}}|^2}{2\sigma_t^2} \right) \exp \left( -\frac{C}{2\sigma_t^2} \right) \tag{65}$$

By integrating over $x_{t-1}$ and using the fact that the integral of a normalized Gaussian density is 1, we obtain the expression for $Z_t(x_t, z; A_t)$:

$$Z_t(x_t, z; A_t) = \exp \left( -\frac{C}{2\sigma_t^2} \right) \int \frac{1}{(2\pi\sigma_t^2)^{d/2}} \exp \left( -\frac{|x_{t-1} - \mu_{\text{weighted}}|^2}{2\sigma_t^2} \right) dx_{t-1} = \exp \left( -\frac{C}{2\sigma_t^2} \right) \tag{66}$$

Further, $C$ can be simplified as

$$C = A_t\mu_c^T\mu_c + (1-A_t)\mu_u^T\mu_u - (A_t\mu_c + (1-A_t)\mu_u)^T(A_t\mu_c + (1-A_t)\mu_u) \tag{67}$$

$$= A_t(1-A_t)\mu_c^T\mu_c + A_t(1-A_t)\mu_u^T\mu_u - 2A_t(1-A_t)\mu_c^T\mu_u \tag{68}$$

$$= A_t(1-A_t)|\mu_c - \mu_u|^2 \tag{69}$$

This gives us the final closed-form expression:

$$-\sum_{t=1}^{T}\mathbb{E}_{q(x_{0:T},z)}\Big[\log Z_t(x_t,z;A_t)\Big] = \sum_{t=1}^{T}\mathbb{E}_{q(x_{0:T},z)}\frac{A_t(1-A_t)}{2\sigma_t^2}|\mu_c - \mu_u|^2 \tag{70}$$

$$= \sum_{t=1}^{T}\frac{A_t(1-A_t)}{2\sigma_t^2}\mathbb{E}_{q(x_{0:T},z)}|\mu_\theta^c(x_t,z,t) - \mu_\theta^u(x_t,t)|^2 \tag{71}$$

$\square$

According to Equation (41), the log-normalization term quantifies the discrepancy between the conditional and unconditional model predictions, scaled by the variance and the product of $A_t$ and $(1-A_t)$. Since the first KL divergence term already encourages extracting useful information from $z$ to aid estimation, we can interpret the log-normalization term as a stop-gradient applied to the first conditional term $\mu_\theta^c(x_t,z,t)$. As such, it acts as an *inherent regularization mechanism* that encourages alignment between the two. This implicitly promotes the model to extract from $x_t$ the aspects most informative about $z$, thereby improving the unconditional estimation.

### C.1.5  Derivation of Equation (7)

In the multi-latent representation setting, the single latent variable $z$ in Proposition 1 and Equation (4) can be replaced by the collection of latent variables $\mathbf{z}^L$. In particular, the third KL divergence term—which corresponds to the generation of $\mathbf{z}^L$—can be structured sequentially in the order $1, 2, \ldots, L$. Formally, this is expressed as:

$$D_{\mathrm{KL}}(q(\mathbf{z}^L|x_0)||p_\theta(\mathbf{z}^L|x_t)) = \mathbb{E}_{q(\mathbf{z}^L|x_0)}\left[\log\frac{q(\mathbf{z}^L|x_0)}{p_\theta(\mathbf{z}^L|x_t)}\right] \tag{72}$$

$$= \mathbb{E}_{q(\mathbf{z}^L|x_0)}\left[\log\frac{\prod_{l=1}^{L}q(z_l|x_0)}{\prod_{l=1}^{L}p_\theta(z_l|x_t,z_{<l})}\right] \tag{73}$$

$$= \mathbb{E}_{q(\mathbf{z}^L|x_0)}\left[\sum_{l=1}^{L}\log\frac{q(z_l|x_0)}{p_\theta(z_l|x_t,z_{<l})}\right] \tag{74}$$

$$= \sum_{l=1}^{L}\mathbb{E}_{q(z_{<l}|x_0)}\mathbb{E}_{q(z_l|x_0)}\left[\log\frac{q(z_l|x_0)}{p_\theta(z_l|x_t,z_{<l})}\right] \tag{75}$$

$$= \sum_{l=1}^{L}\mathbb{E}_{q(z_{<l}|x_0)}\Big[D_{\mathrm{KL}}(q(z_l|x_0)||p_\theta(z_l|x_t,z_{<l}))\Big] \tag{76}$$

### C.1.6  Further Justifications of Section 2.3

**Justification of the Distribution Approximation.**  In Section 2.3, we address the training-inference discrepancy arising from the lack of ground-truth representation $z$ during inference by approximating the hybrid distribution $\tilde{p}_\theta^{\mathrm{linear}}(x_{t-1}|x_t,z;A_t)$—associated with linear cumulative weights—with $p_\theta(x_{t-1}|x_t,\hat{\mu}_t^z)$ for both training and inference. Here, $\hat{\mu}_t^z$ denotes the mean of the current best estimate $p_\theta(z|x_t)$. We contend that this approximation enhances representation guidance as the process denoises, which is consistent with the linear weight schedule. We provide addition justifications to such approximation in this subsection.

We first examine the behavior of the hybrid model as described in Proposition 3.

**Proposition 3** (Score Function of the Hybrid Distribution). *The score function of the marginal distribution $\tilde{p}_\theta(x_{t-1}|z; A_t)$ implied by the hybrid reverse process $\tilde{p}_\theta(x_{t-1}|x_t, z; A_t)$ defined in Equation* (5) *satisfies:*

$$\nabla_{x_{t-1}} \log \tilde{p}_\theta(x_{t-1}|z; A_t) = A_t \nabla_{x_{t-1}} \log p_\theta(x_{t-1}|z) + (1 - A_t)\nabla_{x_{t-1}} \log p_\theta(x_{t-1}) \qquad (77)$$

*where $p_\theta(x_{t-1}|z)$ and $p_\theta(x_{t-1})$ are the marginal distributions implied by the conditional and unconditional reverse processes, $p_\theta(x_{t-1}|x_t, z)$ and $p_\theta(x_{t-1}|x_t)$, respectively.*

*Proof.* For any reverse process $p(x_{t-1}|x_t)$ and its implied marginal $p(x_{t-1})$, the following relation holds under the fixed forward process $q(x_t|x_{t-1})$ based on the Bayes' rule:

$$p(x_{t-1}|x_t) = q(x_t|x_{t-1})p(x_{t-1})/p(x_t) \qquad (78)$$

By taking the logarithm and the gradient with respect to $x_{t-1}$ and rearranging, we have:

$$\nabla_{x_{t-1}} \log p(x_{t-1}) = -\nabla_{x_{t-1}} \log q(x_t|x_{t-1}) + \nabla_{x_{t-1}} \log p(x_{t-1}|x_t) \qquad (79)$$

Applying this to different reverse processes, including the unconditional (i.e., $p_\theta(x_{t-1}|x_t)$), conditional (i.e., $p_\theta(x_{t-1}|x_t, z)$), and the hybrid (i.e., $\tilde{p}_\theta(x_{t-1}|x_t, z; A_t)$) ones, we have

$$\nabla_{x_{t-1}} \log p_\theta(x_{t-1}) = -\nabla_{x_{t-1}} \log q(x_t|x_{t-1}) + \nabla_{x_{t-1}} \log p_\theta(x_{t-1}|x_t) \qquad (80)$$
$$\nabla_{x_{t-1}} \log p_\theta(x_{t-1}|z) = -\nabla_{x_{t-1}} \log q(x_t|x_{t-1}) + \nabla_{x_{t-1}} \log p_\theta(x_{t-1}|x_t, z) \qquad (81)$$
$$\nabla_{x_{t-1}} \log \tilde{p}_\theta(x_{t-1}|z; A_t) = -\nabla_{x_{t-1}} \log q(x_t|x_{t-1}) + \nabla_{x_{t-1}} \log \tilde{p}_\theta(x_{t-1}|x_t, z; A_t) \qquad (82)$$

Further, the gradient of the log-hybrid reverse process can be written as

$$\nabla_{x_{t-1}} \log \tilde{p}_\theta(x_{t-1}|x_t, z; A_t) = \nabla_{x_{t-1}} \log \left( \frac{1}{Z_t(x_t, z)} p_\theta^{A_t}(x_{t-1}|x_t, z) p_\theta^{1-A_t}(x_{t-1}|x_t) \right) \qquad (83)$$

$$= A_t \nabla_{x_{t-1}} \log p_\theta(x_{t-1}|x_t, z) + (1 - A_t)\nabla_{x_{t-1}} \log p_\theta(x_{t-1}|x_t) \qquad (84)$$

Therefore,

$$\nabla_{x_{t-1}} \log \tilde{p}_\theta(x_{t-1}|z; A_t) = A_t(-\nabla_{x_{t-1}} \log q(x_t|x_{t-1}) + \nabla_{x_{t-1}} \log p_\theta(x_{t-1}|x_t, z)) \qquad (85)$$
$$+ (1 - A_t)(-\nabla_{x_{t-1}} \log q(x_t|x_{t-1}) + \nabla_{x_{t-1}} \log p_\theta(x_{t-1}|x_t)) \qquad (86)$$
$$= A_t \nabla_{x_{t-1}} \log p_\theta(x_{t-1}|z) + (1 - A_t)\nabla_{x_{t-1}} \log p_\theta(x_{t-1}) \qquad (87)$$

$\square$

Proposition 3 shows that the hybrid distribution's score function interpolates between those of the conditional and unconditional models according to the weight $A_t$, which decreases over time. Intuitively, at earlier (less noisy) timesteps, the model places more emphasis on the conditional path (leveraging $z$), while at later timesteps, it increasingly relies on the unconditional model. In practice, by leveraging $p_\theta(x_{t-1}|x_t, \hat{\mu}_t^z)$, we approximate the hybrid score $\nabla_{x_{t-1}} \log \tilde{p}_\theta(x_{t-1}|z; A_t)$ using the tractable score $\nabla_{x_{t-1}} \log p_\theta(x_{t-1}|\hat{\mu}_t^z)$, where $\hat{\mu}_t^z$ is predicted during inference.

Observe that, by Bayes' theorem, the marginal distributions admit the following factorizations:

$$p_\theta^{A_t}(x_{t-1}|z) p_\theta^{1-A_t}(x_{t-1}) = p_\theta(x_{t-1}) p_\theta^{A_t}(z|x_{t-1})/p_\theta^{A_t}(z) \qquad (88)$$
$$p_\theta(x_{t-1}|\hat{\mu}_t^z) = p_\theta(x_{t-1}) p_\theta(\hat{\mu}_t^z|x_{t-1})/p_\theta(\hat{\mu}_t^z) \qquad (89)$$

Taking the gradients of their logarithms, we obtain the following decompositions:

$$\nabla_{x_{t-1}} \log \tilde{p}_\theta(x_{t-1}|z; A_t) = \nabla_{x_{t-1}} \log p_\theta(x_{t-1}) + A_t \nabla_{x_{t-1}} \log p_\theta(z|x_{t-1}) \qquad (90)$$
$$\nabla_{x_{t-1}} \log p_\theta(x_{t-1}|\hat{\mu}_t^z) = \nabla_{x_{t-1}} \log p_\theta(x_{t-1}) + \nabla_{x_{t-1}} \log p_\theta(\hat{\mu}_t^z|x_{t-1}) \qquad (91)$$

These expressions highlight that $\nabla_{x_{t-1}} \log p_\theta(\hat{\mu}_t^z|x_{t-1})$ serves as a practical surrogate for the intractable term $A_t \nabla_{x_{t-1}} \log p_\theta(z|x_{t-1})$. Note that the property of the approximation naturally varies

with $t$. With a smaller $t$, the estimation $\hat{\mu}_t^z = \mathbb{E}_{p_\theta(z|x_t)}[z] \approx \mathbb{E}_{q(z|x_t)}[z]$ is closer to the true latent $z \sim q(z|x_0)$, since less noise has been injected into the input following the Markov structure $z - x_0 - x_t$. As a result, the information content in $\hat{\mu}_t^z$ is more aligned with $z$ when $t$ is small, which matches the increasing influence of the conditional component via $A_t$.

This time-dependent behavior facilitates a smooth interpolation between conditional and unconditional estimation, closely resembling the mechanism of classifier-free guidance [19]. Instead of explicitly sampling from both modes, our approach achieves this interpolation implicitly, using the predicted latent representation $\hat{\mu}_t^z$. This built-in mechanism ensures coherence between the training objective and the inference procedure.

**KL Divergence under the von Mises-Fisher (vMF) Distribution.** The third term in $\mathcal{L}_{\text{VB}}^z$ (Equation (4)) represents the KL divergence between the predicted distribution $p_\theta(z|x_t)$ and the reference distribution $q(z|x_0)$. This measure of discrepancy can be expressed in a tractable form by making suitable assumptions about the distributions. In particular, if both $p_\theta(z|x_t)$ and $q(z|x_0)$ follow von Mises-Fisher (vMF) distributions–i.e., $p_\theta(z|x_t) \sim \text{vMF}(\hat{\mu}_t^z, \kappa)$ and $q(z|x_0) \sim \text{vMF}(\mu_z, \kappa)$, where $\kappa$ is a concentration parameter controlling uncertainty–the KL divergence between these distributions is:

$$D_{\text{KL}}(q(z|x_0)||p_\theta(z|x_t)) = \kappa A_p(\kappa) - \kappa A_p(\kappa)\mu_z^T \hat{\mu}_t^z \tag{92}$$

where $A_p(\kappa) := I_{p/2}(\kappa)/I_{p/2-1}(\kappa)$, and $I_v$ denotes the modified Bessel function of the first kind at order $v$.

Thus, minimizing the KL divergence amounts to maximizing the cosine similarity between the model's predicted mean direction $\hat{\mu}_t^z$ and the ground truth mean $\mu_z$, up to a scaling factor determined by $\kappa$. In practice, this objective is modulated by the representation alignment weight $\beta(n)$ (Equation (9)), which controls the relative emphasis placed on the representation generation component.

## C.2 Provable Distribution Approximation with Representation Aligned Diffusion Models

In this subsection, we quantitatively analyze how representation alignment can help to improve the sampling quality of diffusion models. We provide the result for representation-aligned DDPM in Theorem 1.

**Theorem 1.** *Consider the random variable $x \in \mathbb{R}^d$ with ground truth distribution $x \sim q(x) := q(x_0)$. Assume that the second moment $m$ of $x$ is bounded as $m^2 := \mathbb{E}_{q(x)}[\|x - \bar{x}\|^2] < \infty$, where $\bar{x} := \mathbb{E}_{q(x)}[x]$, and that the score $\nabla \log q(x_t)$ is $L$-Lipschitz for all $t$. For a practically trained diffusion model parameterized by $\theta$, the score estimation error at timestep $t$ is bounded by $\epsilon_{t,\theta}$, i.e., $\mathbb{E}_{q_t(x_t)}[\|s_\theta(x_t, t, \hat{z}_{t,\theta}) - \nabla \log q_t(x_t)\|^2] \leq \epsilon_{t,\theta}^2$. Denote the step size as $h := T/N$, where $T$ is the total diffusion time and $N$ is the number of discretization steps, and assume that $h \preceq 1/L$. Denote $k$-dim isotropic Gaussian distribution as $\gamma^k$. Then the following holds,*

$$\text{TV}(p_\theta, q) \preceq \underbrace{\sqrt{\text{KL}(q||\gamma^d)}\exp(-T)}_{\text{convergence of forward process}} + \underbrace{(L\sqrt{dh} + Lmh)\sqrt{T}}_{\text{discretization error}} + \underbrace{\sum_{k=1}^N \epsilon_{kh,\theta}\sqrt{h}}_{\text{score estimation error}} \tag{93}$$

*Proof.* Recall the notation that $p_\theta(x) := p_0$ is the distribution predicted by the denoising network $\theta$ starting from Gaussian noise $\gamma^d$. Consider the two measures over the path space: (i) $Q_T^\leftarrow$, where $(X_t)_{t \in [0,T]}$ follows the law of the reverse process; (ii) $P_T^{q_T}$, under which $(X_t)_{t \in [0,T]}$ has the law of the score matching generative model algorithm initialized at $q_T$ instead of $\gamma^d$, where $q_T$ is the distribution at timestep $T$ in the forward process starting from $q_0 := q$. Denote the end distribution of $P_T^{q_T}$ as $p_0^{q_T}$, we have the following inequality:

$$\text{TV}(p_0, q) \leq \text{TV}(p_0, p_0^{q_T}) + \text{TV}(p_0^{q_T}, q_0) \tag{94}$$

The convergence of the OU process in KL divergence [8] bounds the first term,

$$\text{TV}(p_0, p_0^{q_T}) \leq \text{TV}(\gamma^d, q_T) \leq \sqrt{\text{KL}(q||\gamma^d)}\exp(-T) \tag{95}$$

The second term essentially consists of score estimation error and discretization error. We next show the following holds,

$$\text{TV}(p_0^{q_T}, q_0)^2 \le \text{KL}(q_0||p_0^{q_T}) \preceq h(\sum_{k=1}^{T/h} \epsilon_{kh,\theta}^2) + (L^2 dh + L^2 m^2 h^2)T \tag{96}$$

We start proving Equation (96) by proving

$$\sum_{k=1}^{N} \mathbb{E}_{Q_T^{\leftarrow}} \int_{(k-1)h}^{kh} ||s_\theta^{(kh)}(x_{kh}, kh, \hat{z}_{t,\theta}) - \nabla \ln q_t(x_t)||^2 \mathrm{d}t \preceq (\sum_{k=1}^{N} \epsilon_{kh,\theta}^2)h + (L^2 dh + L^2 m^2 h^2)T \tag{97}$$

For $t \in [(k-1)h, kh]$, we decompose

$$\mathbb{E}_{Q_T^{\leftarrow}}[||s_\theta^{(kh)}(x_{kh}, kh, \hat{z}_{kh,\theta}) - \nabla \ln q_t(x_t)||^2] \tag{98}$$

$$\preceq \mathbb{E}_{Q_T^{\leftarrow}}[||s_\theta^{(kh)}(x_{kh}, kh, \hat{z}_{kh,\theta}) - \nabla q_{kh}(x_{kh})||^2] + \mathbb{E}_{Q_T^{\leftarrow}}[||\nabla q_{kh}(x_{kh}) - \nabla q_t(x_{kh})||^2] \tag{99}$$

$$+ \mathbb{E}_{Q_T^{\leftarrow}}[||\nabla q_t(x_{kh}) - \nabla q_t(x_t)||^2] \tag{100}$$

$$\preceq \epsilon_{kh,\theta}^2 + \mathbb{E}_{Q_T^{\leftarrow}}\left[||\nabla \ln \frac{q_{kh}}{q_t}(x_{kh})||^2\right] + L^2 \mathbb{E}_{Q_T^{\leftarrow}}[||x_{kh} - x_t||^2] \tag{101}$$

Utilizing Lemma 16 from [8], we bound the second term as follows,

$$||\nabla \ln \frac{q_{kh}}{q_t}(x_{kh})||^2 \preceq L^2 dh + L^2 h^2 ||x_{kh}||^2 + (L^2 + 1)h^2 ||\nabla \ln q_t(x_{kh})||^2 \tag{102}$$

where

$$||\nabla \ln q_t(x_{kh})||^2 \preceq ||\nabla \ln q_t(x_t)||^2 + ||\nabla \ln q_t(x_{kh}) - \nabla \ln q_t(x_t)||^2 \tag{103}$$

$$\preceq ||\nabla \ln q_t(x_t)||^2 + L^2 ||x_{kh} - x_t||^2 \tag{104}$$

Combining all the terms, we obtain

$$\mathbb{E}_{Q_T^{\leftarrow}}[||s_\theta^{(kh)}(x_{kh}, kh, \hat{z}_{kh,\theta}) - \nabla \ln q_t(x_t)||^2] \tag{105}$$

$$\preceq \epsilon_{kh,\theta}^2 + L^2 dh + L^2 h^2 \mathbb{E}_{q_0}[||x_{kh}||^2] + L^2 h^2 \mathbb{E}_{q_0}[||\nabla \ln q_t(x_t)||^2] + L^2 \mathbb{E}_{q_0}[||x_{kh} - x_t||^2] \tag{106}$$

$$\preceq \epsilon_{kh,\theta}^2 + L^2 dh + L^2 h^2(d + m^2) + L^3 dh^2 + L^2(m^2 h^2 + dh) \tag{107}$$

$$\preceq \epsilon_{kh,\theta}^2 + L^2 dh + L^2 h^2 m^2 \tag{108}$$

Similarly to [8], we can apply the Girsanov theorem and complete stochastic integration using the properties of Brownian motions and local martingales. Note that $q_0$ is the end of the reverse SDE, by the lower semicontinuity of the KL divergence and the data-processing inequality [5], we take the limit and obtain

$$\text{KL}(q_0||p_0^{q_T}) \preceq (\sum_{k=1}^{N} \epsilon_{kh,\theta}^2)\frac{T}{N} + (L^2 dh + L^2 h^2 m^2)T \tag{109}$$

where we recall that $N := \frac{T}{h}$. We finally conclude with Pinsker's inequality ($\text{TV}^2 \le \text{KL}$) and Cauchy-Schwarz inequality.

$\square$

Theorem 1 is the first theoretical result that gives a strict TV distance bound between the distribution generated by representation-aligned diffusion models and the ground truth. This is a tighter bound compared to [9] thanks to the benefit of external representations in improving the score estimation. In particular, a common bound of the score estimation error $\epsilon_{\text{score}}^2$ assumed at all timesteps $t$ in [9]. In our case, however, since the model latents $\hat{z}_{t,\theta}$ is a good estimation of the ground truth representation $z$, it provides nonzero information about the score, resulting in a smaller error. To interpret this, note that the score $\nabla \log q(x_t)|_{x_t=\tilde{x}}$ is not a function only of the individual sample $\tilde{x}$, but a function of the entire distribution $q(x_t)$ that is related to other samples. External representations, taking advantage of their pretraining tasks and additional training corpus, are able to capture part of the distributional information. Therefore, we can express the score function as $\nabla \log q(x_t)|_{x_t=\tilde{x}} = \psi(\tilde{x}, \hat{z}_t(\tilde{x}), r(x_t)|_{x_t=\tilde{x}})$,

where $\hat{z}_t(\tilde{x})$ is the estimated representation of $\tilde{x}$ and $r(x_t)|_{x_t=\tilde{x}}$ denotes the remaining information not captured by either $\tilde{x}$ or $\hat{z}_t(\tilde{x})$. When training a DDPM, the diffusion loss only helps in predicting $\tilde{x}$ (due to the equivalence between $x$-parameterization and $\epsilon$-parameterization), while incorporating the representation alignment loss additionally facilitates predicting $\hat{z}_t(\tilde{x})$, which is crucial in improving score estimation. Therefore, by choosing suitable representations, a representation-aligned diffusion model is capable of predicting more accurate scores with the help of estimated representations $\hat{z}_{t,\theta}$, i.e., $\mathbb{E}_{q(x_t)}[\|s_\theta(x_t, t, \hat{z}_{t,\theta}) - \nabla \log q(x_t)\|^2] \leq \epsilon_{t,\theta}^2 \leq \epsilon_{\text{score}}^2$. Since predicting representations at smaller $t$ is easier, we can arguably expect that the improvement is even larger at small timesteps $t$.

Another advantage of pretrained representations is their rich and easy-to-obtained information for downstream predictions, which is particularly beneficial in conditional generation. In class-conditioned image generation, for example, class labels can be easily inferred from the DINO representations (e.g., through linear probing), thus aligning diffusion models with these representations provides strong guidance towards the conditions.

## D   Additional Method Details

### D.1   Further Justifications for the Joint Training Curriculum

As discussed in Section 3.2, improved estimates of the representations $\hat{\mu}_t^z$ naturally enhance the modeling of the representation-conditioned distribution in the first term of $\mathcal{L}_{\text{VB}}^z$. One simple and natural way to fulfill this goal is a two-stage training paradigm, where we first initialize the model by aligning with pretrained representations, and then optimize through the original diffusion loss. This approach is equivalent to optimizing the first and the last term in Equation (4) independently, offering the model better initializations.

However, we argue that two-staged training is suboptimal for the following reasons: (i) the sudden change in loss paradigm leads to gradient direction discontinuities in the optimization dynamics, causing the learned representations to collapse (as observed in our experiments); (ii) the representation spaces of the pretrained encoder and the diffusion model (particularly, the VAE space for latent diffusion) are disjoint, while for diffusion models both representation learning and distribution generation are coherent tasks learned by a single neural network, requiring coherent training flow. Therefore, only conducting representation learning may extract the non-informative features for downstream generation while neglecting the informative part. In other words, simply optimizing the third term in Equation (4) can result in a suboptimal starting point to optimize the first term, suggesting that diffusion and representation generation loss cannot be fully decoupled.

The above reasoning highlights the criticality of joint training all loss terms in Equation (4) with the special emphasis on representation learning in the early training period. Hence, as described in Equation (9), we propose a single-stage training curriculum where the diffusion loss weight progressively increases from zero, with a fixed or decaying representation alignment loss.

### D.2   Additional Information for Molecule Representations

Generative models over molecules should respect permutational and geometric symmetries, such as $S(N)$ symmetry group for 2D graphs and (special) Euclidean group $E(3)$ or $SE(3)$ for 3D point clouds. To preserve symmetries, the denoising network should be equivariant with respect to the group transformations. Equivariance ensures that the resulting diffusion process produces invariant distributions. Analogously, molecule representations can be constructed to be either equivariant or invariant. For example, node (atom) level and edge (bond) level features can be made $S(N) \times E(3)$- or $S(N) \times SE(3)$-equivariant, while graph or molecular level features remain invariant.

To ensure these properties, we follow [35] where graph-level invariant representations were shown to improve generation quality with minimal additional computational cost. In our work, we adopt diffusion and flow matching models with equivariant GNN backbones to maintain permutation and geometric equivariances. Additionally, we incorporate lightweight, semantic graph-level invariant representations derived from pretrained $SE(3)$ invariant molecular graph encoders. While equivariant node and edge level representations can also be incorporated as guidance for diffusion model training, we leave this as future directions.

# E  Experimental Details

## E.1  Image Generation

For continuous diffusion on Euclidean data, we experiment on class-conditional image generation using the ImageNet [12] dataset at a resolution of $256 \times 256$, following the experimental setup in REPA [61].

**Evaluation Metrics.**  We generate 50,000 samples for evaluation, using the SDE Euler-Maruyama sampler with the number of diffusion timesteps fixed at 250 for all experiments. Consistent with SiT [40] and REPA [61], we set the last step of the SDE sampler to 0.04. We adopt the evaluation protocol and reference batches of ADM [13], utilizing their official implementation[6]. For evaluation, we use NVIDIA A100 80GB or A6000 50GB GPUs, and enable tf32 precision to accelerate generation as in REPA. We use the following metrics to measure the generation performance:

- **Fréchet Inception Distance (FID)** [18]. FID is a standard metric for evaluating image generation, measuring the similarity between the distributions of real and generated images. It does so by calculating the distance between their feature embeddings obtained from the Inception-v3 network[51], assuming that both sets of features follow multivariate Gaussian distributions.

- **Inception Score (IS)** [47]. Also leveraging the Inception-v3 network, IS evaluates the quality and diversity of generated samples by measuring the KL-divergence between the original label distribution and the distribution of logits after softmax normalization.

- **Precision and Recall** [29]. These metrics follow classic definitions, with precision indicating the proportion of realistic images, and recall reflecting the fraction of training data manifold covered by generated data.

**Baselines.**  We compare REED with recent diffusion-based generation baselines that utilize different inputs and architectural designs:

- *Pixel diffusion models.* ADM [13] enhances U-Net-based diffusion models and introduces classifier-guided sampling to balance quality and diversity. VDM++ [27] presents an adaptive noise schedule to improve training efficiency. Simple diffusion [22] focuses on high-resolution image synthesis by simplifying both noise schedules and architectures. CDM [21] proposes a cascaded approach, training several diffusion models at increasing resolutions, similar to progressiveGAN [26], to generate high-fidelity images via successive super-resolution stages.

- *Latent diffusion with U-Net.* LDM [46] proposes latent diffusion models that model image distributions in a compressed latent space, significantly improving training efficiency while maintaining generation quality.

- *Latent diffusion using transformer and U-Net hybrids.* U-ViT-H/2 [4] introduces a ViT-based latent diffusion model with U-Net-style skip connections. DiffiT [16] improves transformer-based diffusion with a time-dependent multi-head self-attention mechanism. MDTv2-XL/2 [15] proposes an asymmetric encoder-decoder framework, utilizing U-Net-like long shortcuts in the encoder and dense input shortcuts in the decoder for efficient diffusion transformer training.

- *Latent diffusion with transformers.* MaskDiT [63] employs an asymmetric encoder-decoder for efficient transformer-based diffusion, training the model using an auxiliary mask reconstruction task similar to MAE [17]. SD-DiT [65] builds on MaskDiT by adding a self-supervised discrimination objective with a momentum encoder. DiT [43] introduces a pure transformer architecture for diffusion models, utilizing AdaIN-zero modules. SiT [40] systematically investigates training efficiency by shifting from diffusion to continuous flow-based modeling. SiT with REPA [61] further accelerates training by enforcing representation alignment between intermediate layers and DINOv2 [42] patch representations.

**Image Captioning.**  For each ImageNet image, we generate a synthetic caption using Qwen2-VL [55] 2B, using the prompt "Describe this image." and the raw image as the input. We adopt the official conversation template of Qwen2-VL-2B to arrange the text and image information. We set the number of maximum new tokens to be 200, which leads to concise and informative captions. We

---

[6]https://github.com/openai/guided-diffusion/tree/main/evaluations.

provide a few examples of the generated captions and the corresponding image and class labels in Figure 3.

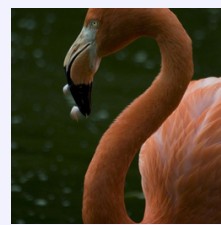

**Class:** Flamingo

**Caption:** *The image depicts a flamingo with its head turned slightly to the side. The flamingo has a long, curved neck and a pinkish-orange plumage. Its beak is black with a white tip, and it appears to be holding something in its beak, possibly a small object or food. The background is a dark green, possibly indicating a water body or a forested area. The flamingo's posture suggests it might be in the midst of a feeding or grooming behavior.*

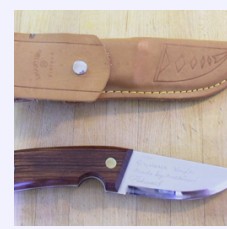

**Class:** Scabbard

**Caption:** *The image shows a knife with a brown leather sheath. The knife has a blade that appears to be made of a polished metal, possibly stainless steel, and it has a handle that is also made of a similar material, likely wood. The sheath is made of brown leather and has a decorative design on it. The knife and sheath are placed on a flat surface, possibly a table or a desk.*

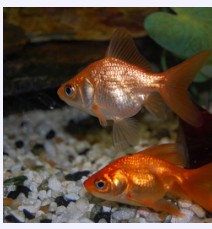

**Class:** Goldfish

**Caption:** *The image shows two goldfish in an aquarium. The goldfish on the left is facing the camera, with its body oriented towards the viewer. The goldfish on the right is facing away from the camera, with its body oriented towards the other goldfish. Both goldfish have a shiny, reflective surface, indicating they are likely well-maintained. The aquarium has a gravel bottom with some small rocks and a few large green plants. The lighting in the aquarium creates a warm, inviting atmosphere.*

Figure 3: Examples of images with their ground-truth class labels and the generated captions.

**VLM Embeddings.** Both the generated caption and the image are then provided as input to the Qwen2-VL 7B model, which contains 28 joint vision-language transformer layers. We extract representations from the 15th layer of Qwen2-VL 7B and average across all image and text tokens to obtain a unified vision-language embedding of dimension 3584. We then align the averaged SiT image patch latents from an intermediate SiT layer, after passing through an MLP projector, with this joint VLM embedding, using a cosine similarity loss (Equation (8)). We choose VLM embeddings over unimodal pretrained text embeddings because VLMs offer a shared embedding space that is already partially aligned between modalities, facilitating better transfer and alignment across image and text representations.

It is noteworthy that we utilize VLMs with different sizes for captioning and representation generation. In particular, we use the smaller Qwen2-VL 2B model to generate the synthetic caption since it is already capable of generating captions with high quality and contains sufficiently useful information of the given image. Yet it is much faster than the 7B model in the generation speed. In contrast, during representation generation, the representations can be obtained in one forward pass for a given image-caption pair, which is much less time-consuming than the sequential inference in the captioning step. Therefore, we opt to use the embeddings of the 7B model, which has stronger capability to understand the input content and is thus able to generate more powerful multimodal representations that accelerate diffusion model training. This is also validated empirically in Table 12.

**Model Architecture and Further Implementation Details.** We adopt the same SiT architectures as REPA and SiT, specifically SiT-B/2, SiT-L/2, and SiT-XL/2, which have 12, 24, and 28 layers, hidden dimensions of 768, 1024, and 1152, and 12, 16, and 16 attention heads, respectively. Latent vectors are pre-computed from raw images using the stable diffusion VAE [46], resulting in $32 \times 32 \times 4$

latent vectors that serve as input to SiT. For decoding, we use the `stabilityai/sd-vae-ft-ema` decoder to reconstruct images from latent vectors.

For image SSL embeddings, we use DINOv2-B and set the alignment coefficient $\lambda_x$ in Equation (8) to 1.0. For VLM embeddings, we use the aforementioned VLM representation with the coefficient $\lambda_y$ set to 0.5. Alignment depth is as follows: for SiT-B, the 4th layer is aligned with the image embedding and the 8th layer with the VLM embedding; for SiT-L and SiT-XL, the 8th layer is aligned with the image embedding and the 16th layer with the VLM embedding. The projection heads for alignment are implemented as 3-layer MLPs with SiLU activations [14]. Regarding the training curriculum (Equation (9)), we keep $\beta(n)$ fixed at 0.5 and increase $\alpha(n)$ linearly during the first 50K iterations, after which it remains at 1.0 for the remainder of training.

For optimization, we use AdamW [39] with a constant learning rate of $1 \times 10^{-4}$, parameters $(\beta_1, \beta_2) = (0.9, 0.999)$, and no weight decay. The batch size is set to 256. To accelerate training, we use mixed-precision (fp16) and apply gradient clipping with a threshold of 1.0.

**Computing Resources.**    Our experiments are conducted using 8 NVIDIA A100 80GB GPUs.

**Analysis on Computational Cost.**    We pre-calculate and store all VLM representations before diffusion model training. As a result, the additional computational overhead during REED training compared to REPA is minimal, limited only to a lightweight MLP projection head applied to the cached VLM representations. This overhead is negligible in practice, thus the actual training time for both REPA and REED is essentially the same. For example, training SiT-XL/2 for 1M iterations (i.e. 200 epochs) on 8×A100 GPUs takes approximately 180 hours, and 4M iterations (i.e. 800 epochs) takes about 720 hours for both methods.

In contrast, the one-time computational cost for generating captions and VLM representations is minor relative to the overall diffusion training cost: on 8×A100 GPUs, captioning requires about 7.5 hours, and VLM representation extraction about 1.3 hours. Therefore, 1M iterations (200 epochs) of REED, togther with captioning and representation generation, takes 188.8 hours in total, much smaller than 4M iterations (800 epochs) of REPA (720 hours). Furthermore, the pre-calculated representations can be reused across all experimental settings and training runs and the time cost is fixed, irrelevant to the number of training iterations. Therefore, the computational overhead introduced by REED is negligible compared to the total training cost.

On the other hand, following REPA, the DINOv2 representations are calculated during diffusion training and not precomputed, which introduces additional computational cost to both REED and REPA in comparison to the vanilla SiT training. However, the specific DINOv2 variant we use, DINOv2-B/14, has only 86M parameters, significantly smaller than the diffusion backbones used in our experiments, from SiT-B/2 (130M params) to SiT-XL/2 (675M params). Also, the representation calculation does not require gradient backpropagation. Therefore, the additional overhead of calculating DINOv2 representations per training step is minimal compared to the overall training cost. For SiT-XL/2, it takes about only 1.05× training time per step compared with the vanilla SiT training, which is far outweighed by the 23.3× acceleration in training steps enabled by our method. Overall, the FLOPS and memory consumption of REED is almost the same as training a vanilla diffusion model.

## E.2   Protein Sequence Generation

For sequence data, we investigate discrete flow models in the context of protein inverse folding, where the goal is to generate protein sequences conditioned on a given 3D backbone structure.

**Dataset and Evaluation Metrics.**    We train the inverse folding models on the PDB training set from [11] and evaluate on the corresponding PDB test set, using the same data splitting. We focus on single-chain proteins with fewer than 256 residues, resulting in 13,753 training proteins and 811 test proteins after filtering. For each test structure, each model generates one protein sequence. Both ProteinMPNN [11] and the discrete flow model use a temperature of 0.1, while the discrete flow model uses 500 diffusion timesteps.

We assess inverse folding performance sequence recovery rate, self-consistency RMSD (scRMSD), and pLDDT from ESMFold [36]. Details for each metric are provided below:

- **Sequence Recovery Rate.** This metric measures the proportion of residues in the generated sequence that match the ground truth. The average is computed at the token level, dividing the number of correctly predicted residues by the total number of residues across all test sequences.

- **scRMSD.** Self-consistency RMSD evaluates how closely the generated sequence, when folded by ESMFold [36], matches the target backbone structure. We report the median scRMSD across all test proteins in Table 5. Since an scRMSD below 2Å is typically considered indicative of successful inverse folding [7, 41], we also report the fraction of test cases with scRMSD less than 2Å.

- **pLDDT.** To further assess sequence quality, we use pLDDT scores from ESMFold predictions on generated sequences. pLDDT reflects the confidence of structure predictions, providing insight into the plausibility of the generated protein sequences. We report the median pLDDT across all test proteins, as well as the percentage of predictions with pLDDT above 80, following standard practice [57].

**Baselines.** We benchmark REED against ProteinMPNN [11], the de facto approach for inverse folding, as well as discrete flow models trained without REED. Following [41, 54], the discrete flow baselines use the Multiflow [7] objective and the ProteinMPNN [11] architecture. All models are trained on our curated dataset as described above, employing a backbone noise of 0.1 and constructing the graph using the 30 nearest neighbors. All other hyperparameters are kept consistent with ProteinMPNN (using 3 encoder and 3 decoder layers). In the original ProteinMPNN, edge representations are fixed and node embeddings are not learnable. To enable better representation alignment with REED, we modify the architecture to allow edge representations to be updated and node features to be learnable. For ablation, we also report the performance of discrete flow models with these architectural adaptations but without REED in Table 13.

**Protein Folding and AF3 Representations.** We utilize AlphaFold3 [1] to generate auxiliary structural information from target protein sequences. For faster inference, we omit the use of Multiple Sequence Alignment (MSA). Structural representations from the penultimate layer of AlphaFold3's diffusion head are extracted as latents, while amino acid representations are obtained from the single and pair embeddings of the last Pairformer layer (corresponding to node and edge features, respectively). When sampling through the diffusion module, we use 10 rounds of recycles and 200 diffusion steps. Given that protein inverse folding is conceptually the reverse of protein folding, we align the layers of our discrete diffusion model with those of the folding model in reverse order. Specifically, the output of the ProteinMPNN encoder, which captures the protein backbone structure, is aligned with the structure representation from AF3. Additionally, after passing through an MLP projector, the output of the decoder's first layer is aligned with the single and pair token embeddings from Pairformer: decoder node embeddings are aligned to single embeddings, and edge embeddings are aligned to pair embeddings.

**Further Implementation Details.** For the training curriculum, we keep $\beta(n)$ fixed at 0.2, while $\alpha(n)$ is linearly increased to 1.0 over the first 50 epochs and then decayed using a cosine schedule. The alignment coefficients $\lambda$s are set to 0.5 for single representations, 2.0 for pair representations, and 1.0 for structure representations. The projection heads for alignment are implemented as 2-layer MLPs with SiLU activations. Optimization is performed using Adam [28] with a constant learning rate of $1 \times 10^{-3}$.

**Computing Resources.** AlphaFold3 representation generation is performed on 8 NVIDIA A100 80GB GPUs, with the extracted embeddings stored for subsequent use. All other experiments are conducted on a single NVIDIA A100 80GB GPU.

### E.3  Molecule Generation

For structured and geometric graph data, we tackle the 3D molecule generation task using equivariant diffusion models, ensuring molecular data symmetries are preserved throughout the process.

**Datasets and Evaluation Metrics.** QM9 [45] and GEOM-DRUG [3] are commonly used benchmarks for unconditional molecule generation. Compared with QM9 where molecule contain approximately 18 atoms on average and up to 9 heavy atoms, GEOM-DRUG is a more challenging and useful dataset of larger, drug-like molecules with an average size of around 44 atoms, which is used to assess our

method. Following SemlaFlow [25], we use the same data splits and discard molecules with more than 72 atoms in the training set (corresponds to about $1\%$ of training data), while keeping validation and test sets unchanged.

We adopt standard benchmark evaluation metrics: *atom stability*, *molecule stability*, *validity*, as well as additional metrics *energy* and *strain*. We also include the number of function evaluations (NFE) to sample one batch of molecules for fair comparison, taking sampling efficiency into account. A thorough description of these metrics is provided as below:

- **Atom Stability.** Fraction of atoms with correct valency.
- **Molecule Stability.** Fraction of molecules in which all atoms are stable.
- **Validity.** Fraction of molecules convertible to valid SMILES strings using RDKit.
- **Energy.** The energy $U(x)$ of a conformation $x$, computed with MMFF94 in RDKit, a commonly used molecular modeling framework [24], where lower values indicate greater physical plausibility.
- **Strain.** Energy difference $U(x) - U(\tilde{x})$ between the generated conformation $x$ and the MMFF94-optimized conformation $\tilde{x}$, with lower values indicating less distortion.

**Baselines.** Recent 3D molecular generators consist of two main types depending on the approaches they obtain bonds: generate atoms only and infer bonds based on coordinates with external rules, or directly generate bonds jointly with atoms. As explained in [25], the latter methods can produce higher-quality samples, including MiDi [53], EQGAT-diff [31], SemlaFlow [25], which we consider as the current state-of-the-art. We only take these advanced 2D&3D methods that directly learn to generate bonds as baselines. In our experiments, we choose SemlaFlow, an $E(3)$-equivariant ODE flow matching, as our base model.

For further context, modeling molecular graph distributions requires careful treatment of both permutational and geometric symmetries. For 2D molecular graphs, the relevant symmetry group is $S(N)$, capturing all permutations of atom indices. For 3D molecular structures, symmetries are governed by the special Euclidean group $SE(3) = SO(3) \ltimes \mathbb{R}^3$, where $SO(3)$ represents rotations and $\mathbb{R}^3$ represents translations. In some cases, the full Euclidean group $E(3) = O(3) \ltimes \mathbb{R}^3$, which also includes reflections, may be considered, but $SE(3)$ is often more practical as it preserves molecular handedness. To maintain these symmetries, the denoising network must be equivariant to the appropriate groups, and the diffusion process must model distributions invariant to these transformations. Accordingly, node-level (atom) and edge-level (bond) features should be permutation equivariant, while graph-level (molecule) features should be permutation invariant. For 3D coordinates, analogously, their node-level embeddings should be $SE(3)$ equivariant, while graph-level features should be $SE(3)$ invariant.

**Pretrained Representation.** For the pretrained encoder, we select the Unimol [64] with $SE(3)$ transformers, and additionally add GEOM-DRUG to its pre-training dataset as described in [35]. We use graph-level invariant representations to align with SemlaFlow intermediate layers. For our 12-layer model, we empirically find that aligning the fourth layer yields the best performance.

**Further Implementation Details.** For the training curriculum, we keep $\beta(n)$ fixed at 1.0, while $\alpha(n)$ is linearly increased to 1.0 over the first 30 epochs and then fixed as a constant for the remaining epochs. The alignment coefficient $\lambda$ is set to 0.2. The projection heads for alignment are implemented as 3-layer MLPs with SiLU activations. We follow all other hyperparameters regarding model, training and evaluation settings in [25].

**Computing Resources.** We train SemlaFlow with REED for 200 epochs on a single NVIDIA A100 80GB GPU. Notably, it requires significantly fewer GPU hours than EQGAT-diff, which is trained for 800 epochs on 4 NVIDIA A100 GPUs.

# F   Additional Experimental Results

## F.1   Image Generation

**Additional Evaluation Results for Table 1.** We present evaluation results across multiple metrics for our method trained with SiT models of various sizes, compared to both the vanilla SiT and REPA

Table 7: Comprehensive evaluation results for comparisons with SiT and REPA across various model sizes. No classifier-free guidance is used. Iter. denotes training iteration.

| Model | #Params | Iter. | FID↓ | IS↑ | Prec.↑ | Rec.↑ |
|---|---|---|---|---|---|---|
| SiT-B/2 | 130M | 400K | 33.0 | 43.7 | 0.53 | 0.63 |
| + REPA | 130M | 50K | 78.2 | 17.1 | 0.33 | 0.48 |
| + REED (ours) | 130M | 50K | 66.0 | 22.9 | 0.37 | 0.51 |
| + REPA | 130M | 100K | 49.5 | 27.5 | 0.46 | 0.59 |
| + REED (ours) | 130M | 100K | 36.8 | 42.5 | 0.51 | 0.61 |
| + REPA | 130M | 200K | 33.2 | 43.7 | 0.54 | 0.63 |
| + REED (ours) | 130M | 200K | 24.6 | 62.2 | 0.58 | 0.64 |
| + REPA | 130M | 400K | 24.4 | 59.9 | 0.59 | 0.65 |
| + REED (ours) | 130M | 400K | 19.5 | 75.9 | 0.61 | 0.65 |
| SiT-L/2 | 458M | 400K | 18.8 | 72.0 | 0.64 | 0.64 |
| + REPA | 458M | 50K | 55.4 | 23.0 | 0.43 | 0.53 |
| + REED (ours) | 458M | 50K | 41.7 | 36.7 | 0.50 | 0.59 |
| + REPA | 458M | 100K | 24.1 | 55.7 | 0.62 | 0.60 |
| + REED (ours) | 458M | 100K | 19.1 | 75.3 | 0.63 | 0.63 |
| + REPA | 458M | 200K | 14.0 | 86.5 | 0.67 | 0.64 |
| + REED (ours) | 458M | 200K | 12.1 | 101.8 | 0.67 | 0.65 |
| + REPA | 458M | 400K | 10.0 | 109.2 | 0.69 | 0.65 |
| + REED (ours) | 458M | 400K | 8.9 | 122.4 | 0.69 | 0.66 |
| SiT-XL/2 | 675M | 7M | 8.3 | 131.7 | 0.68 | 0.67 |
| + REPA | 675M | 50K | 52.3 | 24.3 | 0.45 | 0.53 |
| + REED (ours) | 675M | 50K | 38.0 | 40.4 | 0.52 | 0.59 |
| + REPA | 675M | 100K | 19.4 | 67.4 | 0.64 | 0.61 |
| + REED (ours) | 675M | 100K | 16.0 | 84.4 | 0.65 | 0.63 |
| + REPA | 675M | 200K | 11.1 | 100.4 | 0.69 | 0.64 |
| + REED (ours) | 675M | 200K | 9.8 | 114.7 | 0.68 | 0.65 |
| + REPA | 675M | 400K | 7.9 | 122.6 | 0.70 | 0.65 |
| + REED (ours) | 675M | 400K | 7.3 | 134.5 | 0.70 | 0.66 |
| + REPA | 675M | 4M | 5.9 | 157.8 | 0.70 | 0.69 |
| + REED (ours) | 675M | 1M | 4.7 | 166.1 | 0.72 | 0.66 |

Table 8: Detailed evaluation results for SiT-B/2 models aligned using VLM embeddings from different layers of the Qwen2-VL 7B model. Dep.-I and Dep.-VL denote the alignment depth for image and VLM representations respectively. No classifier-free guidance is used. Iter. denotes training iteration.

| Iter. | Dep.-I | Dep.-VL | VLM Embedding | FID↓ | IS↑ | Prec.↑ | Rec.↑ |
|---|---|---|---|---|---|---|---|
| 200K | 4 | 8 | Layer 0 | 29.9 | 50.6 | 0.56 | 0.64 |
| 200K | 4 | 8 | Layer 1 | 30.2 | 49.6 | 0.55 | 0.64 |
| 200K | 4 | 8 | Layer 15 | **29.4** | 50.8 | **0.56** | 0.64 |
| 200K | 4 | 8 | Layer 27 | 29.5 | **51.2** | 0.56 | **0.65** |

in Table 7. Hyperparameter settings and implementation details are as described in Appendix E.1. REED consistently outperforms both REPA and vanilla SiT across different model sizes, training iterations, and evaluation metrics.

**Results with Different VLM Embeddings.** We evaluate the performance of REED on SiT-B/2 using VLM embeddings extracted from various layers of the Qwen2-VL 7B model, as shown in Table 8. For all experiments, the image (DINOv2) embedding is aligned at the 4th layer and the VLM embedding at the 8th layer, using the same hyperparameters detailed in Appendix E.1. Notably, the best results are achieved when aligning with the 15th layer (out of 28) of Qwen2-VL 7B. Early layers do not provide well-fused vision and text token embeddings, while later layers are more similar to the final VLM output (text) space. Therefore, aligning at a middle layer offers a balanced and robust semantic representation.

Table 9: Detailed evaluation results at different alignment depth on SiT-B/2. Dep.-I and Dep.-VL denote the alignment depth for image and VLM representations respectively. No classifier-free guidance is used. Iter. denotes training iteration.

| Iter. | Dep.-I | Dep.-VL | FID↓ | IS↑ | Prec.↑ | Rec.↑ |
|-------|--------|---------|------|-----|--------|-------|
| 200K | 2 | - | 33.2 | 46.9 | 0.53 | 0.64 |
| 200K | 4 | - | **27.7** | **55.5** | **0.56** | 0.64 |
| 200K | 6 | - | 27.9 | 54.7 | 0.56 | **0.65** |
| 200K | 8 | - | 30.1 | 51.8 | 0.55 | 0.64 |
| 200K | 4 | 2 | 26.5 | 57.7 | 0.57 | 0.64 |
| 200K | 4 | 4 | 25.9 | 59.5 | 0.57 | 0.64 |
| 200K | 4 | 6 | 24.8 | 61.9 | 0.57 | 0.63 |
| 200K | 4 | 8 | **24.6** | **62.2** | **0.58** | 0.64 |
| 200K | 4 | 10 | 26.4 | 59.1 | 0.57 | **0.65** |
| 200K | - | 4 | **45.5** | **33.6** | **0.47** | 0.60 |
| 200K | - | 8 | 45.6 | 32.3 | 0.47 | **0.61** |

**Additional Results for Different Alignment Depths in Table 4.** We present further evaluation results across various alignment depths for both image and VLM representations in Table 9. The observed trends remain consistent across most metrics (except recall): aligning image representations at a shallower layer (i.e., layer 4) and VLM representations at a deeper layer (i.e., layer 8) out of the total 12 layers yields the best performance. This observation is consistent with findings from the Hierarchical VAE literature, where shallower latents tend to capture fine-grained details, while deeper latents encapsulate higher-level semantic information. These results provide empirical support for our theoretical analysis in Section 2.2.

**Empirical Verification of Approximations in Section 2.3.** To assess the validity of the approximations introduced in Section 2.3, we train the model using an alternative objective that directly follows Equation (4) without additional approximations. Specifically, we use the SiT-B/2 model and apply the cosine similarity representation alignment loss (corresponding to the third term in Equation (4)) at the 4th layer. Additionally, we introduce an extra input condition $z$ via cross attention at the start of the 5th layer in SiT. With this architectural adjustment, $p_\theta(x_{t-1}|x_t, z)$ is computed using the ground truth representation as the input condition, while $p_\theta(x_{t-1}|x_t)$ is obtained with a learnable dummy latent vector (representing no conditioning) as the input. The second term is derived according to Equation (41), with its relative coefficient set to 1.0.

The weighted product of two Gaussians with equal variance remains Gaussian, where the mean is the weighted average of the individual means and the variance is unchanged. As a result, under the Gaussian assumption of $p_\theta(x_{t-1}|x_t, z)$ and $p_\theta(x_{t-1}|x_t)$, the distribution $\tilde{p}_\theta(x_{t-1}|x_t, z; A_t)$ in Equation (5) can be expressed in closed form as an aggregation of the conditioned and unconditioned model outputs. This allows the first term in Equation (4) to be calculated as a standard score matching loss but relative to the weighted average of the two model outputs. During inference, since the ground-truth representation $z$ is not available, we sample using the unconditional model $p_\theta(x_{t-1}|x_t)$.

We conduct experiments using DINOv2 representations, keeping all other settings consistent with those used for REPA. The results, presented in Table 10, show that the model trained directly with Equation (4) performs comparably to REPA, thereby empirically supporting the validity of the approximations we made in Section 2.3. Additionally, we observe that the second log-normalization term in the objective remains below 0.04, which is considerably smaller than the denoising loss in the first term (approximately 0.7). This further justifies our decision to exclude it from training, as described in Section 2.3.

**Ablation Study on Different Curriculum Schedules.** To measure the effect of the curriculum schedule between representation learning and diffusion training, we conduct ablations on different configurations for the schedules $\alpha(n)$ and $\beta(n)$, including varying the phase-in period for $\alpha(n)$ and applying a cosine decay to $\beta(n)$. As shown in Table 11, all curriculum variants of $\alpha(n)$ outperform the constant schedule, with 50K iterations chosen as the default for all image generation experiments. Notably, the choice of $\beta(n)$ schedule does not substantially affect generation performance.

Table 10: Results of SiT-B/2 model trained with Equation (4) without approximation (w/o Approx.) and with DINOv2 representations compared with REPA. No classifier-free guidance is used. Iter. denotes training iteration.

| Model (SiT-B/2) | #Params | Iter. | FID↓ | IS↑ | Prec.↑ | Rec.↑ |
|---|---|---|---|---|---|---|
| + REPA | 130M | 200K | 33.2 | 43.7 | 0.54 | 0.63 |
| + Equation (4) w/o Approx. | 130M | 200K | 34.7 | 41.9 | 0.54 | 0.62 |

Table 11: Ablation study on different curriculum schedules. We report results on SiT-B/2 training for 200K iterations. No classifier-free guidance is used.

| Model | $\alpha(n)$ | $\beta(n)$ | FID↓ | IS↑ | Prec.↑ | Rec.↑ |
|---|---|---|---|---|---|---|
| + REPA | - | - | 33.2 | 43.7 | 0.54 | 0.63 |
| + REED | Constant | Constant | 29.4 | 50.8 | 0.56 | 0.64 |
| + REED | Linear increase in 50K iterations | Constant | **24.6** | **62.2** | 0.58 | **0.64** |
| + REED | Linear increase in 25K iterations | Constant | 25.2 | 60.7 | 0.58 | 0.64 |
| + REED | Linear increase in 75K iterations | Constant | 24.7 | 62.2 | 0.58 | 0.64 |
| + REED | Linear increase in 50K iterations | Cosine decay in 200K iterations | 24.7 | 61.4 | **0.59** | 0.63 |

Adaptively or automatically learning these schedules could offer further benefits, for instance by adjusting the weighting based on the difficulty of individual samples or the training stage. For example, the training of samples with easy-to-extract representations could transition more quickly to a higher diffusion loss weight, while more challenging samples could undergo a longer phase-in period. However, designing and integrating such adaptive schedules presents additional challenges and is beyond the current scope of this work. We leave this as a promising direction for future research.

**Ablation Study on Different Multimodal Representations.** We conduct additional ablations on the multimodal representations obtained from different pretrained models. We compare REED (using Qwen2-VL 7B representations) to versions using either a more lightweight model (i.e., Qwen2-VL 2B, OpenCLIP[7], and SigLIP [62]) or intentionally noisier features (i.e., Qwen2-VL 7B with Gaussian noise). All other settings were kept consistent.

The results are shown in Table 12. Using lightweight or less accurate VLM representations still consistently leads to improved performance compared to the model without aligning with multi-modal representations. This demonstrates the robustness of our method to less accurate or noisy representations and suggests that REED retains its advantages even when ideal pretrained features are unavailable. Nevertheless, higher-quality representations (i.e., Qwen2-VL 7B) yield the strongest results, indicating that the quality of pretrained representations remains an important factor for maximizing performance.

However, utilizing text representations from OpenCLIP or SigLIP yields little improvement or even slightly worse performance than the setting without multimodal representations. This can be attributed to the fact that OpenCLIP and SigLIP are trained with a contrastive objective on short and concise captions and are primarily optimized for downstream classification tasks without capturing fine-grained semantic details needed for high-quality image generation. As a result, the information content in their text embeddings may be insufficient to fully support image generation. In comparison, VLMs are trained on a diverse set of vision-language tasks and have strong ability of vision-language understanding and generation. They are also better at bridging two modalities, and consequently, have better aligned visual and textual representations compared to OpenCLIP and SigLIP. These enable VLMs to generate richer and more relevant semantic representations for guiding image generation.

Moreover, from the computational perspective, OpenCLIP and SigLIP requires a higher computational cost than Qwen2-VL 2B. Since neither of them can generate text captions, the captioning step using Qwen2-VL 2B is always necessary, and we can save the Qwen2-VL 2B embeddings along the way to avoid redundant computation in the representation generation stage.

---

[7]The text encoder in `https://huggingface.co/stabilityai/stable-diffusion-xl-base-1.0`.

Table 12: Ablation study on using different multimodal representations for REED. We report results on SiT-B/2 training for 200K iterations. No classifier-free guidance is used.

| Multimodal Representation | FID↓ | IS↑ | Prec.↑ | Rec.↑ |
|---|---|---|---|---|
| - | 27.7 | 55.5 | 0.56 | 0.64 |
| Qwen2-VL 7B | **24.6** | **62.2** | **0.58** | 0.64 |
| Qwen2-VL 2B | 26.3 | 59.9 | 0.57 | **0.65** |
| Qwen2-VL 7B + noise (scale=0.1) | 25.0 | 61.8 | 0.58 | 0.64 |
| OpenCLIP | 29.0 | 55.8 | 0.55 | 0.64 |
| SigLIP | 27.6 | 57.6 | 0.56 | 0.64 |

## F.2 Protein Sequence Generation

**Ablation Study on Architectural Modifications.** As described in Appendix E.2, we modify the original ProteinMPNN architecture to enhance representation alignment with REED by enabling updates to edge representations and making node features learnable. For the ablation study, we evaluate discrete flow models incorporating these architectural changes but without applying REED, as shown in Table 13. The results are comparable to those of the original DiscreteDiff model, suggesting that the observed performance improvements stem from the REED algorithm itself.

Table 13: Ablation study on the base discrete flow model with the same architectural modifications as DiscreteDiff+REED. The performance remains similar to the original DiscreteDiff model, indicating that the improvements are attributable to REED itself.

| Methods | Epochs | Seq. Recovery(%) ↑ | scRMSD ↓ | %(scRMSD<2) ↑ | pLDDT ↑ | %(pLDDT>80)(%) ↑ |
|---|---|---|---|---|---|---|
| DiscreteDiff w/ | 50 | 38.04 | 2.034 | 49.32 | 81.36 | 58.39 |
| Architectural | 100 | 40.60 | 1.794 | 53.42 | 82.82 | 68.32 |
| Modifications | 150 | 41.35 | 1.817 | 54.78 | 83.00 | 68.20 |

**Ablation Study on Aligned Representations.** We present ablation results for various combinations of aligned representations in Table 14, including using only single representations, only pair representations, both single and pair representations, and only structure representations. These are compared against the results obtained when utilizing all three representation types. For each ablation (except "all"), we fix $\beta(n)$ at 0.2 and set the alignment coefficients $\lambda$ to 1.0 for all representations. As shown in Table 14, each representation type independently contributes substantially to model performance, with all outperforming the baseline DiscreteDiff model at each training epoch. Notably, the pair representation yields the largest performance gain, highlighting the importance of capturing amino acid interactions for improved protein structure understanding in the inverse folding task.

Table 14: Ablation studies on different combinations of aligned representations: single, pair, and structure. "all" denotes using all three types of representations.

| Representation | Epochs | Seq. Recovery(%) ↑ | scRMSD ↓ | %(scRMSD<2) ↑ | pLDDT ↑ | %(pLDDT>80)(%) ↑ |
|---|---|---|---|---|---|---|
| single | 50 | 40.35 | 1.790 | 53.91 | 82.76 | 67.21 |
| | 100 | 41.43 | 1.778 | 55.03 | 83.20 | 67.83 |
| | 150 | 42.06 | 1.768 | 55.28 | 83.50 | 71.68 |
| pair | 50 | 41.07 | 1.775 | 55.53 | 83.13 | 68.94 |
| | 100 | 41.92 | 1.781 | 55.03 | 82.89 | 68.70 |
| | 150 | 42.21 | 1.778 | 55.53 | 83.02 | 69.32 |
| single+pair | 50 | 40.62 | 1.806 | 55.28 | 83.12 | 68.32 |
| | 100 | 41.69 | 1.766 | 56.52 | 83.04 | 66.96 |
| | 150 | 41.85 | 1.753 | 54.53 | 83.13 | 69.57 |
| structure | 50 | 40.40 | 1.839 | 54.04 | 82.50 | 64.97 |
| | 100 | 41.47 | 1.774 | 53.91 | 83.08 | 67.58 |
| | 150 | 41.88 | 1.729 | 55.90 | 83.32 | 70.81 |
| all | 50 | 41.06 | 1.788 | 54.37 | 82.74 | 67.41 |
| | 100 | 41.91 | 1.780 | 56.23 | 83.09 | 68.41 |
| | 150 | 42.26 | 1.799 | 54.66 | 83.17 | 69.07 |

# G  Qualitative Results

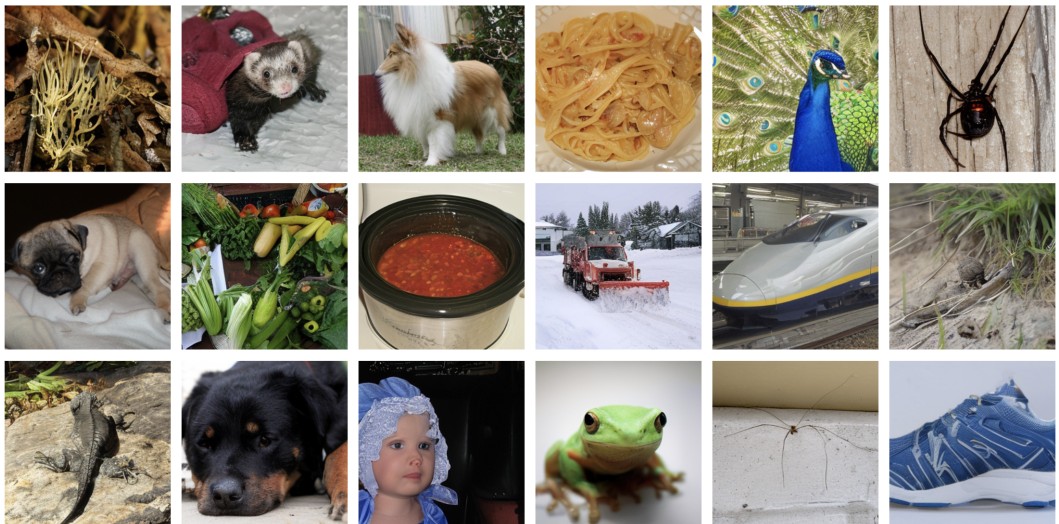

Figure 4: Selected samples on ImageNet $256 \times 256$ generated by the SiT-XL/2+REED model after 1M training iterations. We use classifier-free guidance with $w = 1.275$.

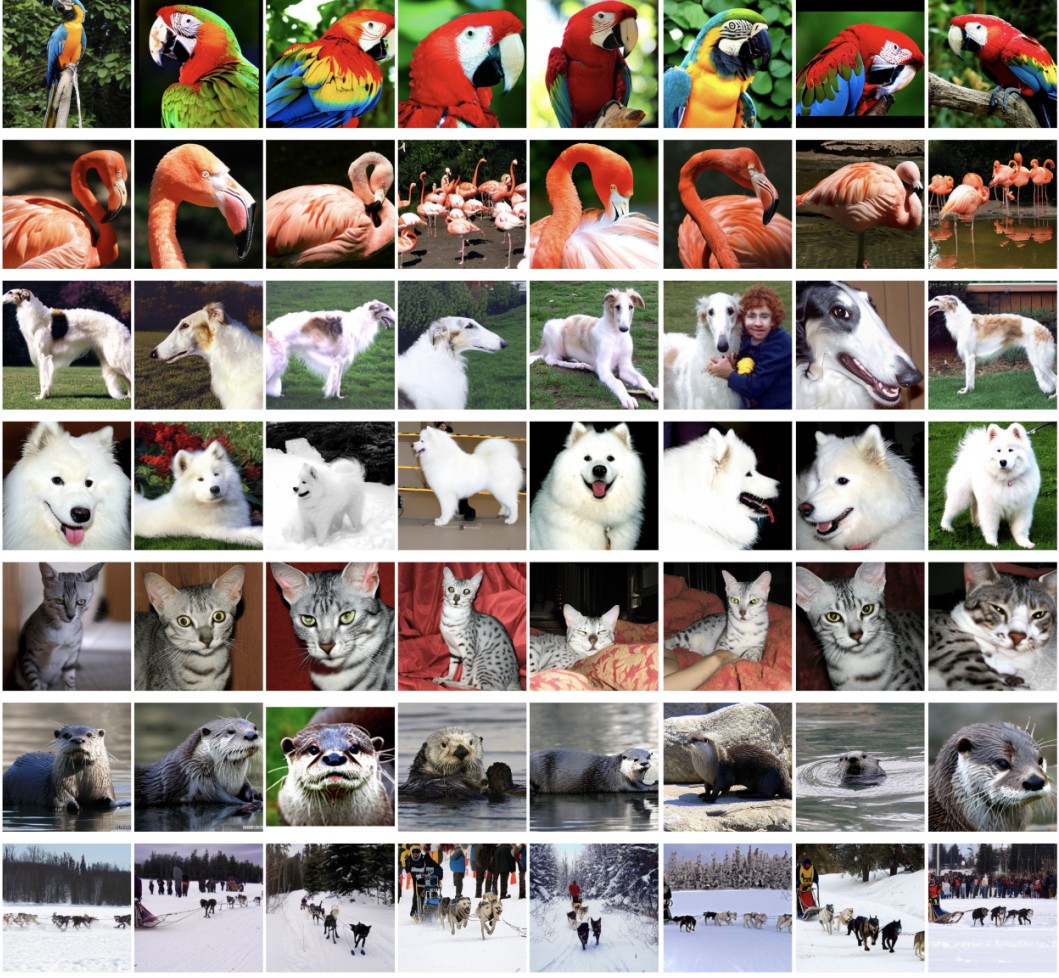

Figure 5: Selected samples on ImageNet $256 \times 256$ generated by the SiT-XL/2+REED model after 1M training iterations. We use classifier-free guidance with $w = 4.0$. Each row corresponds to the same class label. From top to bottom, the class labels are: "macaw" (88), "flamingo" (130), "borzoi, Russian wolfhound" (169), "Samoyed" (258), "Egyptian cat" (285), "otter" (360), and "dogsled" (537), respectively.

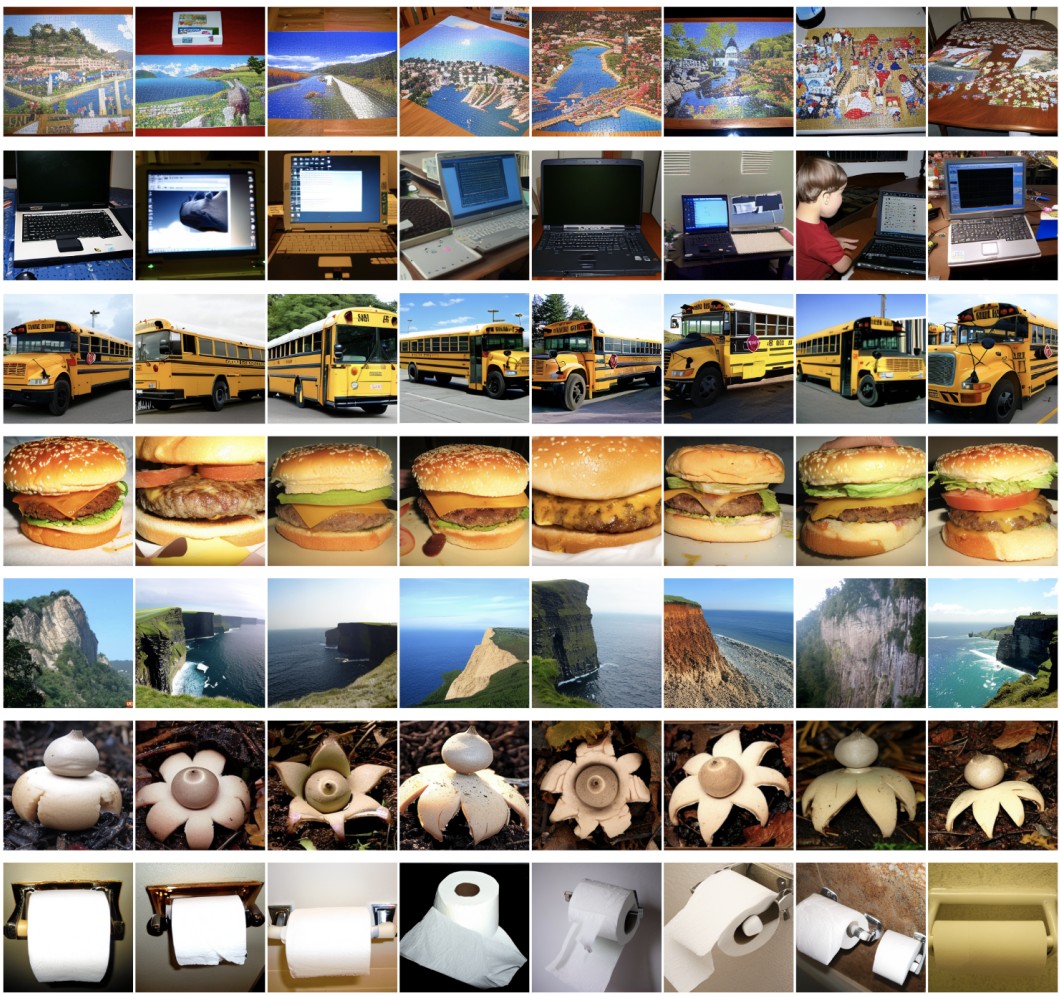

Figure 6: Selected samples on ImageNet $256 \times 256$ generated by the SiT-XL/2+REED model after 1M training iterations. We use classifier-free guidance with $w = 4.0$. Each row corresponds to the same class label. From top to bottom, the class labels are: "jigsaw puzzle" (611), "laptop" (620), "school bus" (779), "cheeseburger" (933), "cliff" (972), "earthstar" (995), and "toilet tissue" (999), respectively.

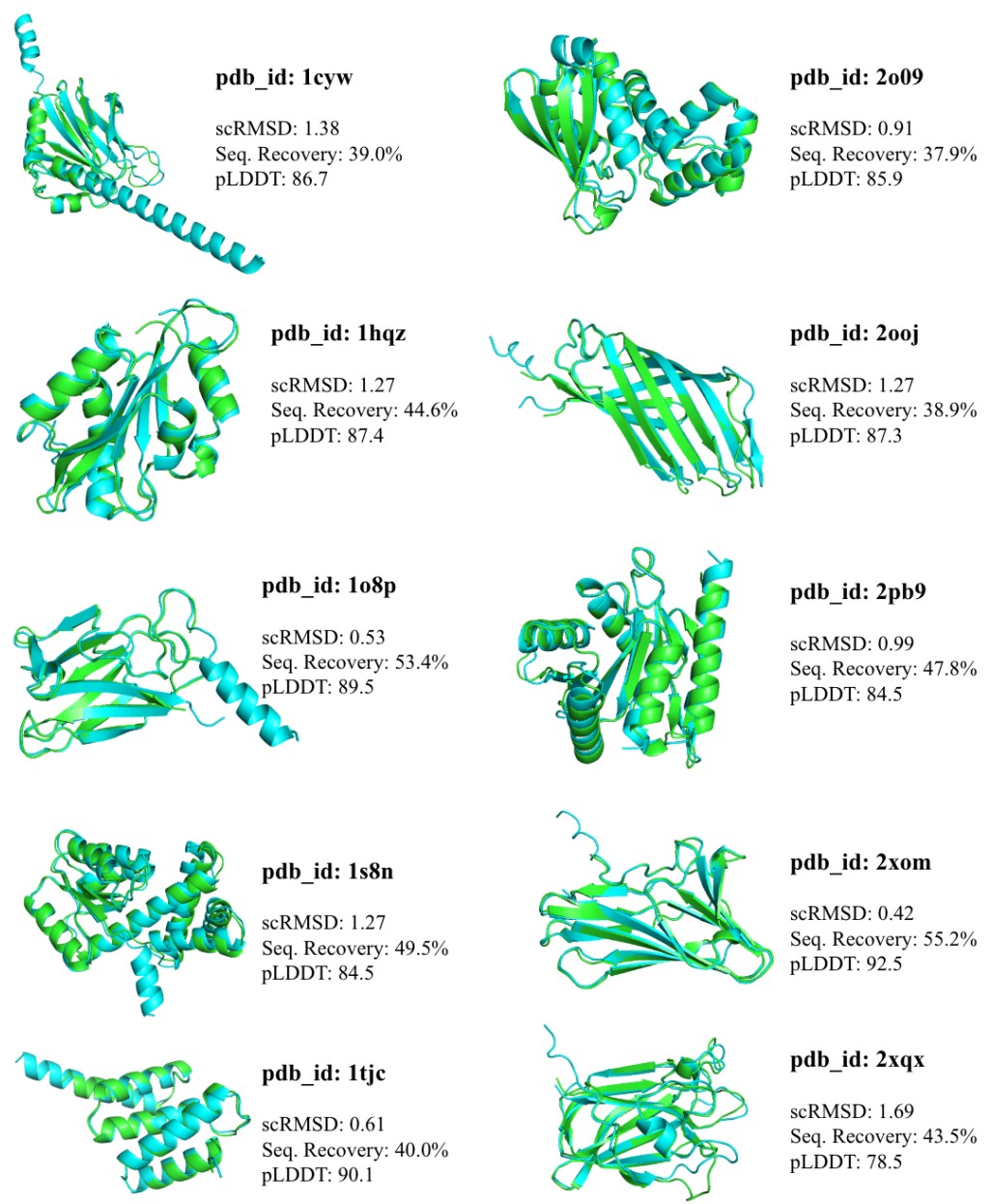

Figure 7: Selected samples on protein inverse folding. Green color denotes the ground truth structure and blue color denotes the generated sequence folded by ESMFold [36].

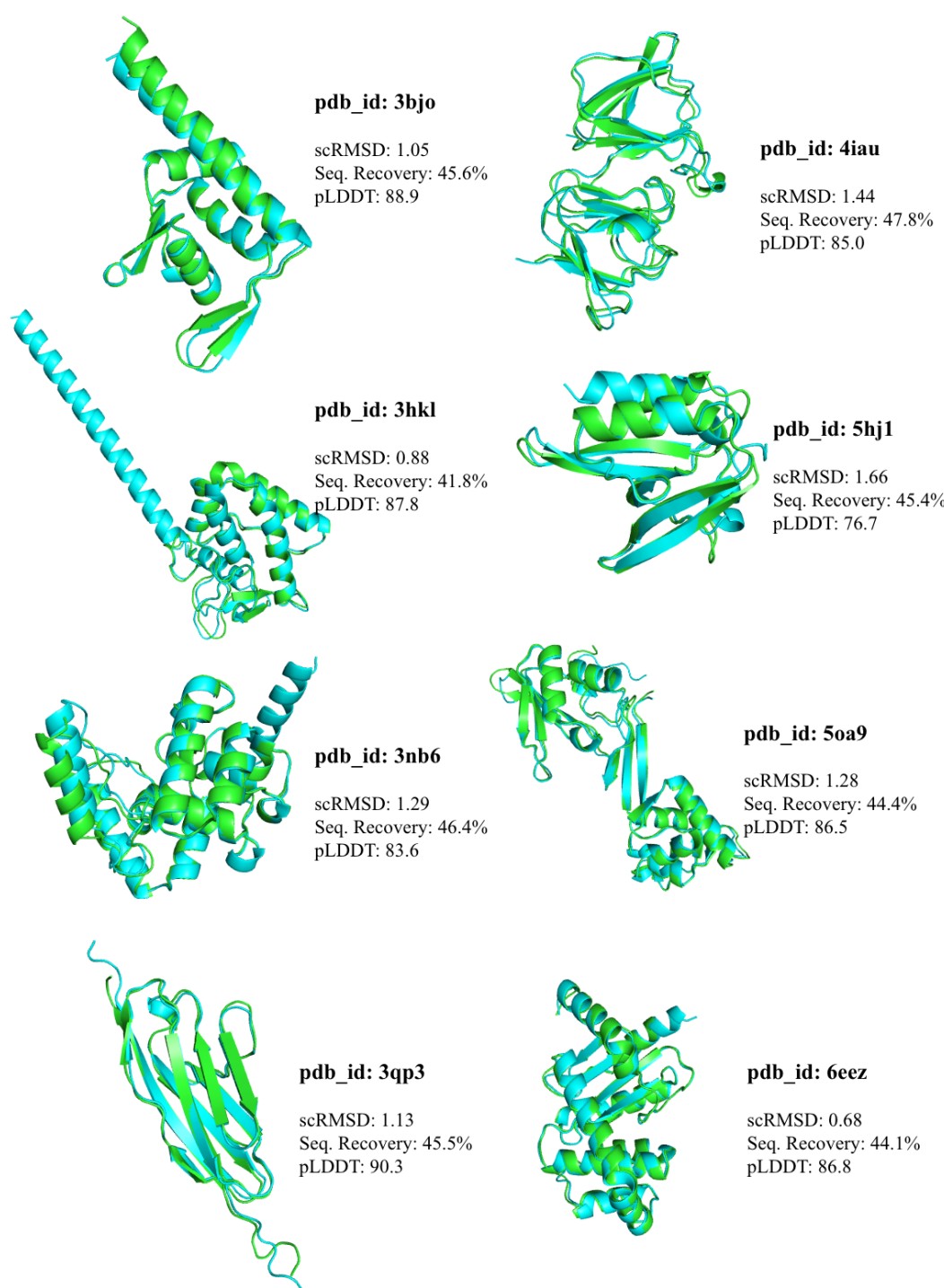

Figure 8: (Continued.) Selected samples on protein inverse folding. Green color denotes the ground truth structure and blue color denotes the generated sequence folded by ESMFold [36].

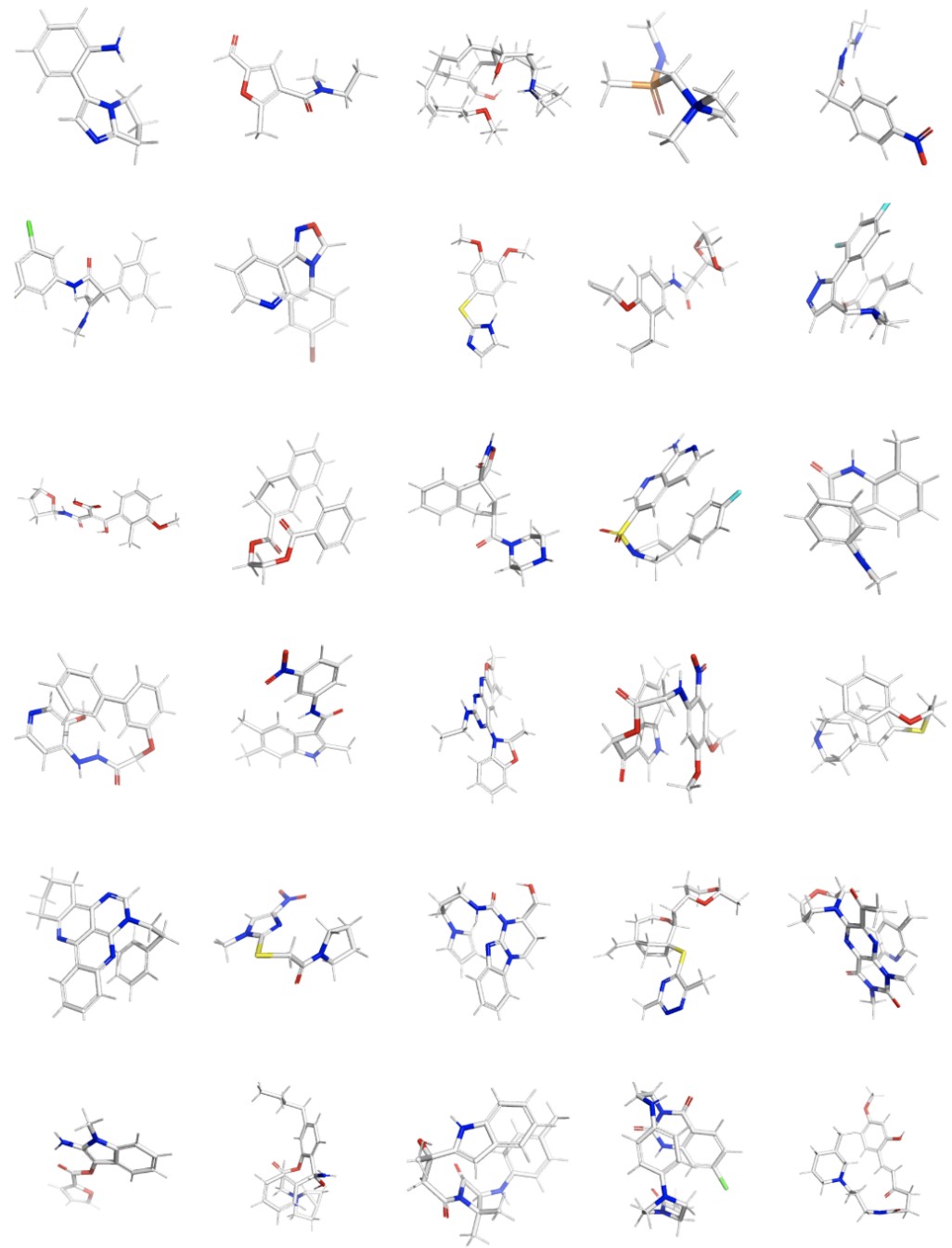

Figure 9: Selected samples on molecule generation.

