# OpenReview forum: "Learning Diffusion Models with Flexible Representation Guidance"
_NeurIPS.cc/2025/Conference — NeurIPS 2025 poster_

### Official Review · Reviewer_WXGL · 2025-06-05

**Clarity:** 2
**Significance:** 3
**Originality:** 3
**Rating:** 5
**Confidence:** 2

**Summary:**

This paper extends REPA, a method originally developed to accelerate diffusion transformer training for image generation, to various modalities. While REPA focuses on aligning intermediate features between DINO and the diffusion transformer, REED further aligns representations from multiple modalities—such as text—during image generation. Additionally, REED is the first to introduce REPA-style regularization terms into protein sequence and molecule generation, demonstrating its applicability beyond vision tasks such as protein sequence generation and molecule generation.

**Questions:**

What would be the effect of using a smaller multimodal model, such as CLIP or SigLIP, instead of Qwen2-VL for image generation? Would REED retain its benefits while becoming more efficient?

**Ethical Concerns:**

["NO or VERY MINOR ethics concerns only"]

**Final Justification:**

The rebuttal addresses my concerns. Pre-computation of representations addresses my concern on computational overhead. The theoretical characterization gives valuable insights. Thus, I raised my score to 5.

**Limitations:**

The section 5 should further discuss the computation overhead by using large external models.

**Paper Formatting Concerns:**

The paper needs a figure illustrating the proposed framework, like the Figure 1 of the REPA paper.

**Quality:**

2

**Strengths And Weaknesses:**

[Strengths]

- Novel generalization of REPA to multimodal representation

- First to apply REPA-like regularization to non-visual generative tasks

- Demonstrates consistent performance improvements over baselines across all tasks

[Weaknesses]

- While REPA was motivated by improving training efficiency, REED uses Qwen2-VL 7B for image generation—a much heavier model than DINO—raising doubts about actual training speed improvements. A comparison of GFLOPS or latency is needed to confirm whether REED's 1M iterations in Table 1 are more efficient than REPA’s 4M, and whether 200 epochs in Table 2 are indeed more efficient than REPA’s 800. Figure 1 should also use GFLOPS or latency on the x-axis instead of training epochs.

- Section 2's theoretical characterization is not novel, and seems unnecessary in this paper. The connection between ELBO and diffusion models is well-known [A], and the effectiveness of timestep weighting [B] has been established in prior work. I feel Sections 2 and 3 are loosely connected.

[A] Kingma et al., Variational Diffusion Models, NeurIPS 2021.

[B] Understanding Diffusion Objectives as the ELBO with Simple Data Augmentation, NeurIPS 2023.

---

> ### Author Rebuttal · Authors · 2025-07-31
>
> We would like to thank the reviewer for their insightful review and their thoughtful feedback. In response to their review, we believe we have addressed their concerns as concretely as possible.
>
> > While REPA was motivated by improving training efficiency, REED uses Qwen2-VL 7B for image generation—a much heavier model than DINO—raising doubts about actual training speed improvements. A comparison of GFLOPS or latency is needed to confirm whether REED's 1M iterations in Table 1 are more efficient than REPA’s 4M, and whether 200 epochs in Table 2 are indeed more efficient than REPA’s 800. Figure 1 should also use GFLOPS or latency on the x-axis instead of training epochs.
>
> We **pre-calculate and store** all VLM representations before diffusion model training. As a result, the additional computational overhead during REED training compared to REPA is minimal, limited only to a lightweight MLP projection head applied to the cached VLM representations. This overhead is negligible in practice, thus the actual training time for both REPA and REED is essentially the same. For example, training SiT-XL/2 for 1M iterations (i.e. 200 epochs) on 8×A100 GPUs takes approximately **180 hours**, and 4M iterations (i.e. 800 epochs) takes about **720 hours** for both methods.
>
> In contrast, the one-time computational cost for generating captions and VLM representations is minor relative to the overall diffusion training cost: on 8×A100 GPUs, caption generation requires about **7.5 hours**, and VLM representation extraction about **1.3 hours**. Therefore, 1M iterations (200 epochs) of REED (+captioning+representation generation) takes 188.8 hours in total, much smaller than 4M iterations (800 epochs) of REPA (720 hours).
>
> Furthermore, the pre-calculated representations can be reused across all experimental settings and training runs and the time cost is fixed, irrelevant to the number of training iterations. Therefore, the computational overhead introduced by REED is negligible compared to the total training cost. While we are unable to include an updated figure due to response restrictions, we will update Figure 1 accordingly and revise the manuscript with these clarifications and discussions.
>
>
> > Section 2's theoretical characterization is not novel, and seems unnecessary in this paper. The connection between ELBO and diffusion models is well-known [A], and the effectiveness of timestep weighting [B] has been established in prior work. I feel Sections 2 and 3 are loosely connected.
> [A] Kingma et al., Variational Diffusion Models, NeurIPS 2021.
> [B] Understanding Diffusion Objectives as the ELBO with Simple Data Augmentation, NeurIPS 2023.
>
>
> Our theoretical characterization is fundamentally different from [A] and [B]. Both [A] and [B] focus on diffusion models **without** external pretrained representations, whereas our work derives, to our knowledge, the first variational bound for diffusion models that incorporate **external representation as additional guidance**. This represents a substantial extension of the existing theory. Moreover, the timestep weighting discussed in our theoretical section specifically addresses how to balance the strength of representation guidance across different timesteps, which is conceptually distinct from the approaches explored in [B].
>
> We believe this new theoretical framework is valuable for several reasons:
> - It offers the first principled explanation of how and why representation-enhanced diffusion models–including prior works such as REPA and RCG–achieve their performance gains.
> - It directly motivates key aspects of our new methods: for example, our use of multimodal representations (including synthetic modalities) is grounded in the analysis of multi-latent representation structures in Section 2.2. Similarly, our curriculum schedule for the alignment and diffusion losses is motivated by the decoupled structure of the variational bound in Section 2.1.
>
> We will clarify these points in the revised manuscript and better highlight the novel connections between our theoretical analysis and the design of our method.
>
> > What would be the effect of using a smaller multimodal model, such as CLIP or SigLIP, instead of Qwen2-VL for image generation? Would REED retain its benefits while becoming more efficient?
>
> As discussed in our response to the first point regarding computational efficiency, we pre-compute and store all captions and corresponding representations before diffusion model training. As a result, the computational cost of generating VLM representations is a one-time, fixed expense and does not scale with the number of training iterations. During diffusion training, the additional overhead from using these representations is negligible. Therefore, the size of the multimodal model used for representation extraction does not significantly impact the overall training efficiency of REED.
>
> In the table below, we present the performance of REED using different multimodal representations with SiT-B/2 trained for 200K iterations and samples generated without classifier-free guidance (CFG). Both Qwen2-VL 7B and Qwen2-VL 2B representations result in substantial improvements over the setting without multimodal representations, while using OpenCLIP text representations yields slightly worse performance. This difference can be attributed to the fact that CLIP is trained with a contrastive objective, which is more suitable for classification tasks and does not provide the fine-grained semantic alignment needed for high-quality image generation. In comparison, VLMs are trained on a diverse set of vision-language tasks and have *strong ability of vision-language understanding and generation*. They are also *better at bridging two modalities*, and consequently, have better aligned visual and textual representations compared to CLIP. These satisfactory features  enable them to generate richer and more relevant semantic representations for guiding image generation.
>
> It is also worth noting that, even from a computational perspective, OpenCLIP actually needs more computational cost than Qwen2-VL 2B, since neither CLIP nor OpenCLIP can generate captions - hence the captioning step using Qwen2-VL 2B is necessary anyways and we can save the Qwen2-VL 2B embeddings along the way to avoid redundant computation.
>
> | Model | Multimodal representation | FID | IS | Pre. | Rec. |
> |-----|-----|-----|-----|-----|-----|
> | REPA | - | 33.2 | 43.7 | 0.54 | 0.63 |
> | REED | - | 27.7 | 55.5 | 0.56 | 0.64 |
> | REED | Qwen2-VL 7B | **24.6** | **62.2** | **0.58** | 0.64 |
> | REED | Qwen2-VL 2B | 26.3 | 59.9 | 0.57 | **0.65** |
> | REED | OpenCLIP | 29.0 | 55.8 | 0.55 | 0.64 |
>
>
> > The paper needs a figure illustrating the proposed framework, like the Figure 1 of the REPA paper.
>
> We thank the reviewer for the suggestion. While we are unable to include an updated figure due to response restrictions, we will update the manuscript with a better figure illustration of the proposed framework in the revised version.

---

> > ### Comment · Reviewer_WXGL · 2025-08-04
> >
> > Thank you for the detailed rebuttal. I raised my score to 5.

---

> > > ### Comment · Reviewer_WXGL · 2025-08-04
> > >
> > > I have one follow-up question regarding the rebuttal. What size of OpenCLIP model was used? Also, is there a particular reason why OpenCLIP was chosen over CLIP and SigLIP?

---

> > > > ### Author Response · Authors · 2025-08-04
> > > > **Response to Follow-up Questions of Reviewer WXGL**
> > > >
> > > > We thank the reviewer for their time and the increased score.
> > > >
> > > > Regarding the follow-up question, since we need a pooled text embedding of the whole caption sentence, we follow the practice in Stable Diffusion XL (SDXL) and use the same OpenCLIP model as SDXL to obtain the pooled text embedding (i.e., `text_encoder_2` from the huggingface repo `stabilityai/stable-diffusion-xl-base-1.0`). This corresponds to OpenCLIP ViT-bigG-14, whose text encoder contains approximately 695 million parameters. Actually, OpenCLIP, CLIP, and SigLIP are pretty similar in terms of their training, usage, and performance, so we choose OpenCLIP as a representative in the experiment.

---

> > > > > ### Comment · Reviewer_WXGL · 2025-08-04
> > > > >
> > > > > Your paper does not seem to include experiments with SDXL. Could you clarify why you chose to follow SDXL? In particular, since SigLIP is reported to outperform OpenCLIP, I was wondering whether using SigLIP might allow the performance to approach that of Qwen2-VL 2B or 7B.

---

> > > > > > ### Author Response · Authors · 2025-08-04
> > > > > > **Response to Reviewer WXGL**
> > > > > >
> > > > > > We chose to follow SDXL as it has demonstrated empirical success in incorporating text embeddings into image diffusion models, although it focuses on text conditioning for text to image generation rather than the multimodal representation alignment for image generation in our paper.
> > > > > >
> > > > > > Following the reviewer’s suggestion, we experimented with the text embedding from SigLIP, where SiT-B/2 is trained for 200K iterations and samples are generated without classifier-free guidance (CFG). The FID is 27.6, much better than using OpenCLIP (FID=29.0) and slightly better than the case without multimodal representations (FID=27.7). We attribute the marginal performance gain to some limitations of the CLIP/SigLIP types of models that they are trained on short and concise captions and are primarily optimized for downstream classification tasks without capturing sufficient details as VLMs. As a result, the information content in their text embeddings may be insufficient to fully support image generation compared to VLMs.

---

> > > > > > > ### Comment · Reviewer_WXGL · 2025-08-05
> > > > > > >
> > > > > > > Thank you for your response. I hope this paper will be accepted.

---

> ### Author Response · Authors · 2025-08-05
>
> Thank you again for your constructive review and strong support of our work! We greatly appreciate your valuable suggestions and will incorporate the discussions into the final version.

---

### Official Review · Reviewer_EPRe · 2025-06-23

**Clarity:** 2
**Significance:** 3
**Originality:** 2
**Rating:** 4
**Confidence:** 3

**Summary:**

This paper presents a systematic framework REED (Representation-Enhanced Elucidation of Diffusion) for incorporating representation guidance into diffusion models. The authors provide theoretical insights into how high-quality pretrained representations can enhance diffusion model training, proposing two novel strategies: (1) integrating multimodal representations with synthetic data, and (2) designing an optimal training curriculum that balances representation learning and data generation. The framework is demonstrated across three domains: image generation, protein sequence generation, and molecule generation, showing good performance and accelerated training compared to existing methods.

**Questions:**

1. The paper mentions using synthetic paired data to address the scarcity of multimodal data. Could the authors provide more details on how the synthetic data is generated? Clarifying this could strengthen the reproducibility of the results.
2. Can you explain why use Qwen2-VL 2B to generate synthetic captions and use representations from Qwen2-VL 7B?
3. Could you provide more details between the theoretical analysis and experiment details?

**Ethical Concerns:**

["NO or VERY MINOR ethics concerns only"]

**Final Justification:**

The author addressed all my concerns, so I raise the score.

**Limitations:**

Yes.

**Paper Formatting Concerns:**

No.

**Quality:**

3

**Strengths And Weaknesses:**

### Strength

1. The paper is technically sound with well-supported claims through both theoretical analysis and extensive experiments.
2. The paper tests the algorithm on three tasks: image generation, protein generation, and molecule generation.
3. REED seems to show better performance compared to baselines.

### Weaknesses

1. Could you provide more details on how the synthetic data is generated by VLM? Such as instruction prompts and caption examples.
2. Experiments use Qwen2-VL 2B to generate synthetic captions and use representations from Qwen2-VL 7B? What’s the purpose to use different VLMs?
3. In Section 3.1, the paper mentions using features from both a primary modality x and a secondary modality y. However, in the experiments, the authors chose to use features from VLMs. While VLMs do incorporate both text and image information, this appears more like a fused feature representation rather than distinct modalities as suggested by the theory. This seems to deviate significantly from the theoretical framework, particularly regarding how hyperparameters control the alignment between two separate features.

    Additionally, if VLMs are used as the aligned features, the key difference from REPA (which uses DINO features) seems to be primarily the choice of features (VLM vs. DINO). Could you elaborate more on the main experimental distinctions between these two works?

4. In Appendix C.1.2, from equation (21) to (22), it seems impossible to derive $p(z|x_0)p(x_0|x_1)=p(z|x_1)p(x_0|x_1,z)$.

---

> ### Author Rebuttal · Authors · 2025-07-31
>
> We thank the reviewer for their time and feedback that helped us improve the work. In response to their review, we believe we have addressed their concerns as concretely as possible. Below, we share our thoughts on the questions asked and will update the manuscript to reflect these clarifications.
>
> > Could you provide more details on how the synthetic data is generated by VLM? Such as instruction prompts and caption examples.
>
> We use Qwen2-VL 2B for image captioning, using the prompt “Describe this image.” and the raw image as the input. We adopt the official conversation template of Qwen2-VL-2B to arrange the text and image information. We set the number of maximum new tokens to be 200, which leads to concise and informative captions. While we are unable to include the corresponding images due to response restrictions, below are a few examples of the generated captions and the corresponding image class labels:
>
> ```
> Class label: flamingo
> Caption:
> The image depicts a flamingo with its head turned slightly to the side. The flamingo has a long, curved neck and a pinkish-orange plumage. Its beak is black with a white tip, and it appears to be holding something in its beak, possibly a small object or food. The background is a dark green, possibly indicating a water body or a forested area. The flamingo's posture suggests it might be in the midst of a feeding or grooming behavior.
> ```
>
> ```
> Class label: scabbard
> Caption:
> The image shows a knife with a brown leather sheath. The knife has a blade that appears to be made of a polished metal, possibly stainless steel, and it has a handle that is also made of a similar material, likely wood. The sheath is made of brown leather and has a decorative design on it. The knife and sheath are placed on a flat surface, possibly a table or a desk.
> ```
>
> ```
> Class label: goldfish
> Caption:
> The image shows two goldfish in an aquarium. The goldfish on the left is facing the camera, with its body oriented towards the viewer. The goldfish on the right is facing away from the camera, with its body oriented towards the other goldfish. Both goldfish have a shiny, reflective surface, indicating they are likely well-maintained. The aquarium has a gravel bottom with some small rocks and a few large green plants. The lighting in the aquarium creates a warm, inviting atmosphere.
> ```
>
> > Experiments use Qwen2-VL 2B to generate synthetic captions and use representations from Qwen2-VL 7B? What’s the purpose to use different VLMs?
>
> We use the smaller Qwen2-VL 2B model to generate the synthetic caption since it is already capable of generating captions with high quality and contains sufficiently useful information of the given image. Yet it is much faster than the 7B model in the generation speed. Since generating captions requires sequential inference and is more time-consuming compared with obtaining the representations in one forward pass for a given image-caption pair, we opt to generate captions with the 2B model and obtain representations with both models.
>
> However, as shown in the table below, while the embeddings of the 2B model help, the 7B embeddings significantly improve the generation quality of REED. The table contains results
> for the SiT-B/2 models trained for 200K iterations and samples generated without classifier-free guidance. For the given image and caption pairs, the 7B model has stronger capability to understand the content and thus is able to generate more powerful multimodal representations that accelerate diffusion model training.
>
> | Model | Multimodal representation | FID | IS | Pre. | Rec. |
> |-----|-----|-----|-----|-----|-----|
> | REPA | - | 33.2 | 43.7 | 0.54 | 0.63 |
> | REED | - | 27.7 | 55.5 | 0.56 | 0.64 |
> | REED | Qwen2-VL 7B | **24.6** | **62.2** | **0.58** | 0.64 |
> | REED | Qwen2-VL 2B | 26.3 | 59.9 | 0.57 | **0.65** |
>
> **In summary, we use Qwen2-VL 2B since it is capable of generating high-quality captions efficiently. We leverage both Qwen2-VL 2B and 7B models for alignment, and the results show that the 7B model leads to superior performance.**
>
> > In Section 3.1, the paper mentions using features from both a primary modality x and a secondary modality y. However, in the experiments, the authors chose to use features from VLMs. While VLMs do incorporate both text and image information, this appears more like a fused feature representation rather than distinct modalities as suggested by the theory. This seems to deviate significantly from the theoretical framework, particularly regarding how hyperparameters control the alignment between two separate features.
> Additionally, if VLMs are used as the aligned features, the key difference from REPA (which uses DINO features) seems to be primarily the choice of features (VLM vs. DINO). Could you elaborate more on the main experimental distinctions between these two works?
>
> We actually use **both the DINOv2 embeddings and the VLM embeddings** in diffusion training following $\mathcal{L}_{\text{repgen}}$ in Equation 8 (Section 3.1). Specifically, for the image generation experiments, we set the coefficient of DINO embedding $\lambda_x$ as 1.0 and the coefficient of VLM embedding $\lambda_y$ as 0.5.
>
> Therefore, compared with REPA, the main experimental distinctions are:
> - We are the first to propose and demonstrate the effectiveness of using other auxiliary modalities to improve generation in a target modality. We believe the idea of **leveraging (synthetic) multimodality to train single-modal generative models** is novel and opens new directions for future research.
> - We introduce a novel curriculum schedule between the representation alignment loss and diffusion loss, which phases in the diffusion loss while maintaining the representation alignment loss from the start.
> - To the best of our knowledge, we are the first to apply representation alignment to diffusion models in the context of protein inverse folding and molecule generation, demonstrating the versatility and effectiveness of our approach across a diverse set of domains and tasks.
>
> Each component of our method designs are well motivated by our theoretical framework and demonstrate significant improvement in the ablation study: as shown in the table with SiT-B/2 model trained for 200K iterations and samples generated without CFG, removing each component of REED leads to inferior performance. We will update the manuscript with more detailed explanations of the method and the experimental settings and these additional results in the revised version.
>
> | Model | FID | IS | Pre. | Rec. |
> |-----|-----|-----|-----|-----|
> | REPA | 33.2 | 43.7 | 0.54 | 0.63 |
> | REED | **24.6** | **62.2** | **0.58** | **0.64** |
> | REED w/o curriculum schedule | 29.4 | 50.8 | 0.56 | 0.64 |
> | REED w/o multimodal representations | 27.7 | 55.5 | 0.56 | 0.64 |
>
>
> > In Appendix C.1.2, from equation (21) to (22), it seems impossible to derive $p(z|x_0)p(x_0|x_1)=p(z|x_1)p(x_0|x_1,z)$
>
> $$p(z|x_1)p(x_0|x_1,z) = p(z, x_0|x_1) = p(x_0|x_1)p(z|x_0,x_1) = p(x_0|x_1)p(z|x_0)$$
> The last equation is due to the Markov chain $x_1-x_0-z$, which indicates $z$ and $x_1$ are conditionally independent given $x_0$.

---

> > ### Author Response · Authors · 2025-08-04
> > **Gentle Reminder to Respond to our Rebuttal**
> >
> > Dear reviewer EPRe,
> >
> > As the discussion period is drawing to a close, we have received positive feedback from the other reviewers. Since we have not yet heard from you, we wanted to kindly request your feedback on our rebuttal. In our latest response, we believe we have addressed your concerns as thoroughly as possible, including providing detailed explanations of synthetic data generation, clarifying the algorithmic implementation and the theoretical formula, and presenting additional experiments with different VLMs.
> >
> > We understand you may have a busy schedule, but if you have any follow-up questions or remaining concerns, please let us know. If our rebuttal has resolved your concerns, we would greatly appreciate it if you could share your feedback and update your score. Thank you for your time.
> >
> > Best regards,
> > Authors

---

> > ### Comment · Reviewer_EPRe · 2025-08-05
> >
> > Thanks for your response, I have no further questions.

---

> ### Author Response · Authors · 2025-08-05
>
> Thank you for your time and response. We are glad that we have addressed your concerns.  We sincerely appreciate your insightful comments and will incorporate the discussions into the final version. We would appreciate if you could update your final justification and rating.

---

### Official Review · Reviewer_edpA · 2025-07-01

**Clarity:** 4
**Significance:** 4
**Originality:** 3
**Rating:** 5
**Confidence:** 2

**Summary:**

Paper introduces aframework for enhancing diffusion models through representation guidance, i.e. aligning the internal representations of diffusion models with pretrained representations like DINO features. The authors derive a variational bound incorporating auxiliary representations and show how a weighting schedule can control when and how these representations guide the denoising process.
two key strategies are proposed: (1) multimodal representation alignment using synthetic paired data (2) training curriculum that phases in the diffusion loss. REED is evaluated across three tasks: image generation, protein inverse folding, and molecule generation. Showing improved performance and faster training compared to prior methods like REPA.

**Questions:**

- Can you provide further empirical analysis supporting why the linear phase-in for the diffusion loss is better than simpler schedules (e.g., constant weighting)? Would adaptive or learned schedules yield further gains?
- What failure modes might occur when the synthetic auxiliary modality is too dissimilar from the primary data? For example, could this lead to misaligned representations that degrade generation quality?
- how does REED perform when using lightweight or less accurate pretrained representations? This is crucial for understanding the method's generalizability
- how sensitive is REED to the differences between the pretrained representation space and the diffusion model's latent space?

**Ethical Concerns:**

["NO or VERY MINOR ethics concerns only"]

**Final Justification:**

I rated this paper highly because it offers a well-motivated and technically solid extension of REPA. Despite some dependence on large pretrained models, the strong empirical results and broad applicability justify an acceptance.

**Limitations:**

The authors have adequately addressed the limitations of their work.

**Quality:**

4

**Strengths And Weaknesses:**

**Strengths**

- The paper is built on a theoretical motivation, deriving a variational bound that incorporates pretrained representations. The experimental section demonstrates clear performance gains.

- The paper is logically organized, moving from theory to methods and then to experiments. Key ideas, such as the use of synthetic multimodal data, are illustrated with figures.

- The paper addresses the important challenge of improving the efficiency and quality of diffusion models. The application to protein and molecular domains shows potential for impact in scientific discovery.


**Weakness**
- The experiments could be stronger with more rigorous ablations to rule out simpler explanations for the performance gains. FOr example,  could the authors report results for a baseline where the full REED model is trained without the proposed curriculum? i.e. with constant weighting for both the diffusion loss and the representation loss?

- The method's success is heavily dependent on extremely large, resource-intensive pretrained models like DINOv2. This reliance is a barrier to wider adoption, and application in resource-constrained settings, potentially limiting the practical impact of the work.

- The work is largely an extension of REPA. While the new components are innovative, the core contribution can be seen as an incremental improvement that combines existing ideas rather than presenting a fundamentally new framework

---

> ### Author Rebuttal · Authors · 2025-07-31
>
> We would like to thank the reviewer for their diligent and insightful review that helped us improve our work and their positive feedback regarding our theoretical framework, presentation, experimental performance and broad applications. In response to their review, we believe we have addressed their concerns as concretely as possible.
>
> > Rigorous ablations to rule out simpler explanations for the performance gains.
>
> This is indeed an essential ablation. We experiment on the class conditional image generation experiment on the SiT-B/2 architecture and report the results for models trained for 200K iterations and not use classifier-free guidance for generation. As shown in the table, removing each component leads to inferior performance than the full REED model, with **FID increasing from 24.6 to 29.4 when removing the curriculum schedule and to 27.7 when removing multimodal representations**. This demonstrates the effectiveness of each component in our proposed REED.
>
> | Model | FID | IS | Pre. | Rec. |
> |-----|-----|-----|-----|-----|
> | REPA | 33.2 | 43.7 | 0.54 | 0.63 |
> | REED | **24.6** | **62.2** | **0.58** | **0.64** |
> | REED w/o curriculum schedule | 29.4 | 50.8 | 0.56 | 0.64 |
> | REED w/o multimodal representations | 27.7 | 55.5 | 0.56 | 0.64 |
>
> > Further empirical analysis supporting why the linear phase-in for the diffusion loss is better than simpler schedules (e.g., constant weighting). Would adaptive or learned schedules yield further gains?
>
> We provide additional empirical results with different curriculum schedules $\\alpha(n)$ and $\\beta(n)$ on the image generation experiment as below with the SiT-B/2 architectures trained for 200K iterations and samples generated without classifier-free guidance. We will include these results in the revised manuscript.
>
> Compared to the REED without the curriculum schedules (i.e. constant schedule), the curriculum schedule leads to significant improvement, reducing the FID from 29.4 to 24.6. This further verifies the effectiveness of the curriculum schedules in accordance with our theoretical insights.
>
> We further examine different configurations for the schedules, including varying the phase-in period for $\\alpha(n)$ and applying a cosine decay to $\\beta(n)$. As shown in the table below, all curriculum variants of $\\alpha(n)$ outperform the constant schedule, with 50K iterations chosen as the default for all image generation experiments. Notably, the choice of $\\beta(n)$ schedule does not substantially affect generation performance.
>
> | Model | $\alpha(n)$ | $\beta(n)$ | FID | IS | Pre. | Rec. |
> |-----|-----|-----|-----|-----|-----|-----|
> | REED | constant | constant | 29.4 | 50.8 | 0.56 | 0.64 |
> | REED | linear increase to 50K iterations | constant | **24.6** | **62.2** | 0.58 | **0.64** |
> | REED | linear increase to 25K iterations | constant | 25.2 | 60.7 | 0.58 | 0.64 |
> | REED | linear increase to 75K iterations | constant | 24.7 | 62.2 | 0.58 | 0.64 |
> | REED | linear increase to 50K iterations | cosine decay in 200K iterations | 24.7 | 61.4 | **0.59** | 0.63 |
>
> We agree that adaptively or automatically learning these schedules could offer further benefits, for instance by adjusting the weighting based on the difficulty of individual samples or the training stage. For example, the training of samples with easy-to-extract representations could transition more quickly to a higher diffusion loss weight, while more challenging samples could undergo a longer phase-in period. However, designing and integrating such adaptive schedules presents additional challenges and is beyond the current scope of this work. We leave this as a promising direction for future research.
>
> > The method's success is heavily dependent on extremely large, resource-intensive pretrained models like DINOv2, potentially limiting the application in resource-constrained settings.
>
> Since it is a core motivation behind representation-enhanced generative methods to leverage powerful external pretrained representations to accelerate diffusion training, this is a common issue for this type of methods including REPA – which also utilizes DINOv2 representations.
>
> However, it is worth noting that the specific DINOv2 variant we use in our experiments, DINOv2-B/14, has only 86M parameters. This is significantly smaller than the diffusion backbones used in our experiments, from SiT-B/2 (130M params) to SiT-XL/2 (675M params).
>
> In terms of computational cost, for SiT-XL/2, the additional overhead of using DINOv2 representations per training step is minimal compared to the overall training cost–it takes about only **1.05x** training time for each step–and is far outweighed by the **23.3x** acceleration in training steps enabled by our method.
>
> Moreover, some pretrained representations (e.g., the VLM representation from Qwen2-VL) can be pre-computed and reused, further reducing the computational cost. Thus, the overall FLOPS and memory consumption is almost the same as training a vanilla diffusion model.
>
> > The work is largely an extension of REPA. While the new components are innovative, the core contribution can be seen as an incremental improvement that combines existing ideas.
>
> Our work has several fundamental and unique contributions compared to REPA and other representation-enhanced generative models:
> - We provide the first theoretical framework for representation-enhanced diffusion model training, which not only explains the success of prior works such as REPA and RCG, but also well motivates our method design and offers guidance for future research in this area.
> - We systematically explore the design space for representation alignment in diffusion models based on our theoretical insights, both theoretically grounded and empirically validated, which lead to significant performance gains.
> - In particular, we are the first to propose and demonstrate the effectiveness of using other auxiliary modalities to improve generation in a target modality. We believe the idea of leveraging (synthetic) multimodality to train single-modal generative models is novel and opens new directions for future research.
> - To the best of our knowledge, we are the first to apply representation alignment to diffusion models in the context of protein inverse folding and molecule generation, demonstrating the versatility and effectiveness of our approach across a diverse set of domains and tasks.
>
> Overall, we believe that our work brings essential new insights into this field and can inspire a series of future research.
>
> > What failure modes might occur when the synthetic auxiliary modality is too dissimilar from the primary data? Could this lead to misaligned representations that degrade generation quality?
>
> Our method is motivated by the relevance of the synthetic auxiliary modality to the generation task of the primary data. Accordingly, we are deliberate in constructing the auxiliary modality, carefully selecting modalities that are highly relevant and leveraging state-of-the-art models for both synthetic modality generation and multimodal representation extraction (see Section 3.3). For example, we use text modalities and vision-language models (VLMs) for image generation, 3D structures and AlphaFold3 for protein inverse folding, and 2D-3D joint molecule graphs with the UniMol model for molecule generation. This ensures that the auxiliary modality provides meaningful complementary information, maximizing its benefit during diffusion training while minimizing the risk of introducing irrelevant noise.
>
> However, if the synthetic modality is too dissimilar from the primary data, there is a risk of misaligned representations. In such cases, the detrimental effects of irrelevant or distracting information could outweigh the benefits of any useful signal, ultimately leading to degraded generation quality. Therefore, careful selection and validation of the auxiliary modality remain essential to avoid these potential failure modes.
>
> > How does REED perform when using lightweight or less accurate pretrained representations?
>
> We conduct additional experiments on the image generation experiment as below with the SiT-B/2 architectures trained for 200K iterations and samples generated without classifier-free guidance. We compare REED (using Qwen2-VL 7B representations) to versions using either a lightweight model (i.e., Qwen2-VL 2B) or intentionally noisier features (i.e., Qwen2-VL 7B with added Gaussian noise). All other settings were kept consistent. The results are presented below:
>
> | Model | Multimodal representation | FID | IS | Pre. | Rec. |
> |-----|-----|-----|-----|-----|-----|
> | REED | - | 27.7 | 55.5 | 0.56 | 0.64 |
> | REED | Qwen2-VL 7B | **24.6** | **62.2** | **0.58** | 0.64 |
> | REED | Qwen2-VL 2B | 26.3 | 59.9 | 0.57 | **0.65** |
> | REED | Qwen2-VL 7B + noise (scale=0.1) | 25.0 | 61.8 | 0.58 | 0.64 |
>
> As shown in this table, using lightweight or less accurate pretrained representations still consistently leads to improved performance compared to the model without aligning with VLM representations. This demonstrates the robustness of our method to less accurate or noisy representations and suggests that REED retains its advantages even when ideal pretrained features are unavailable. Nevertheless, higher-quality representations (i.e., Qwen2-VL 7B) yield the strongest results, indicating that the quality of pretrained representations remains an important factor for maximizing performance.
>
> > How sensitive is REED to the differences between the pretrained representation space and the diffusion model's latent space?
>
> During training, we align the diffusion model’s latent space with both the DINOv2 embedding space and the VLM representation space—each of which constitutes a distinct and independently pretrained representation space. Despite these substantial differences, our empirical results show that REED consistently achieves strong performance. This suggests that REED is rather robust to the differences among embedding spaces.

---

> > ### Comment · Reviewer_edpA · 2025-08-05
> >
> > Thank you for your detailed response to my questions and the additional experiments. I hope you can add them to the final submission.
> > I'm happy to recommend acceptance for your work.

---

> ### Author Response · Authors · 2025-08-05
>
> Thank you again for your insightful review and strong support of our work! We greatly appreciate your valuable suggestions and will incorporate the discussions and additional experimental results into the final version.

---

### Official Review · Reviewer_LKbx · 2025-07-02

**Clarity:** 3
**Significance:** 3
**Originality:** 4
**Rating:** 5
**Confidence:** 4

**Summary:**

This paper presents a unified theoretical framework for integrating pretrained representations into diffusion model training, enabling flexible guidance via multimodal and hierarchical alignment. The proposed method, REED, demonstrates strong improvements in sample quality and training speed across diverse tasks including image generation, protein design, and molecule generation.

**Questions:**

NA

**Ethical Concerns:**

["NO or VERY MINOR ethics concerns only"]

**Final Justification:**

The authors solved my concerns.

**Quality:**

4

**Strengths And Weaknesses:**

**Strengths:**

1. The paper develops a rigorous and general variational framework for understanding and designing representation-guided diffusion models. This subsumes and explains prior works (e.g., REPA, RCG) as special cases, and provides new design insights (e.g., multi-latent structures, flexible scheduling).

2. The proposed approach goes beyond simple alignment with a single pretrained representation. It supports multi-level, multimodal representations (e.g., combining image and text embeddings, or structure and sequence in proteins), and shows empirically that this leads to better controllability and generation quality.

3. The paper motivates and validates a practical curriculum that introduces representation learning loss early and phases in diffusion loss, leading to faster convergence and improved stability.

**Weakness:**

1. While the equivalence to the RCG method is clearly established through the variational bound and decomposed probabilistic framework, the connection to REPA is less explicit and lacks rigorous justification.

2. The paper would benefit from more comprehensive ablation experiments exploring how different choices of the curriculum schedules $\alpha(n)$ and $\beta(n)$ (which control the relative weighting of diffusion and representation alignment losses) affect model performance. Currently, the ablation on these schedules is limited; it remains unclear how sensitive the proposed method is to the selection or annealing strategy of these weights, and whether adaptive or learned schedules might further improve results.

---

> ### Author Rebuttal · Authors · 2025-07-31
>
> We would like to thank the reviewer for their diligent and insightful review that helped us improve our work and their positive feedback regarding our theoretical framework, proposed method and experiments. In response to their review, we believe we have addressed their concerns as concretely as possible.
>
> > While the equivalence to the RCG method is clearly established through the variational bound and decomposed probabilistic framework, the connection to REPA is less explicit and lacks rigorous justification.
>
> We establish the connection between our framework and REPA under the linear cumulative weight setting in Section 2.3 (lines 173–197) and Appendix C.1.6. Specifically, we show that in this setting, $\\mathcal{L}_{\\text{VB}}^z$ (Equation 4) corresponds directly to the REPA loss function. The connection is well-motivated and justified both theoretically and empirically:
> - As detailed in Proposition 3 and Appendix C.1.6, the hybrid distribution $\\tilde{p}_{\theta}^{\\text{linear}}(x\_{t-1}|x\_t,z;A\_t)$ in the first term of $\\mathcal{L}\_{\\text{VB}}^z$ can be approximated by $p\_{\\theta}(x\_{t-1}|x\_t,\\hat{\\mu}\_t^z)$, where $\\hat{\\mu}\_t^z$ is an estimation of $z$ inferred from $x\_t$. ***This approximation renders that the first term is essentially equivalent to the diffusion loss in REPA***, wherein the model extracts information about the pretrained representation $z$ from $x\_t$ in early layers of the transformer and then conducts score estimation based on $x\_t$ and the extracted features (i.e., $\\hat{\\mu}\_t^z$) in later layers.
> - The third term of $\\mathcal{L}\_{\\text{VB}}^z$ aligns the predicted $z$ from $x\_t$ with the ground truth $z$, ***matching the cosine similarity alignment loss in REPA*** under the standard von Mises-Fisher distribution assumption for representations.
> - We omit the log-normalization term due to its equivalent effect to the third term, and further justify this via experiments in Appendix F.1 (line 1195-1217) and Table 9.
>
> Taken together, these points establish a rigorous connection between our theoretical framework and REPA, and provide clear motivation for the novel design choices in REED.
>
> > The paper would benefit from more comprehensive ablation experiments exploring how different choices of the curriculum schedules $\\alpha(n)$ and $\\beta(n)$ (which control the relative weighting of diffusion and representation alignment losses) affect model performance. Currently, the ablation on these schedules is limited; it remains unclear how sensitive the proposed method is to the selection or annealing strategy of these weights, and whether adaptive or learned schedules might further improve results.
>
> This is indeed an essential ablation. We provide additional results on the image generation experiment as below with the SiT-B/2 architecture trained for 200K iterations and samples generated without classifier-free guidance. We will include these results in the revised manuscript.
>
> Our findings clearly show that employing curriculum schedules leads to significant improvements: **using a constant schedule (no curriculum), the FID is 29.4, whereas introducing a curriculum schedule for $\\alpha(n)$ reduces the FID to 24.6**. This provides strong empirical support for the effectiveness of curriculum schedules, in line with our theoretical motivations.
>
> We further examine different configurations for the schedules, including varying the phase-in period for $\alpha(n)$ and applying a cosine decay to $\beta(n)$. As shown in the table below, all curriculum variants of $\alpha(n)$ outperform the constant schedule, with 50K iterations chosen as the default for all image generation experiments. Notably, the choice of $\beta(n)$ schedule does not substantially affect generation performance.
>
> | Model | $\alpha(n)$ | $\beta(n)$ | FID | IS | Pre. | Rec. |
> |-----|-----|-----|-----|-----|-----|-----|
> | REPA | - | - | 33.2 | 43.7 | 0.54 | 0.63 |
> | REED | constant | constant | 29.4 | 50.8 | 0.56 | 0.64 |
> | REED | linear increase to 50K iterations | constant | **24.6** | **62.2** | 0.58 | **0.64** |
> | REED | linear increase to 25K iterations | constant | 25.2 | 60.7 | 0.58 | 0.64 |
> | REED | linear increase to 75K iterations | constant | 24.7 | 62.2 | 0.58 | 0.64 |
> | REED | linear increase to 50K iterations | cosine decay in 200K iterations | 24.7 | 61.4 | **0.59** | 0.63 |
>
> We agree that adaptively or automatically learning these schedules could offer further benefits, for instance by adjusting the weighting based on the difficulty of individual samples or the training stage. For example, the training of samples with easy-to-extract representations could transition more quickly to a higher diffusion loss weight, while more challenging samples could undergo a longer phase-in period. However, designing and integrating such adaptive schedules presents additional challenges and is beyond the current scope of this work. We leave this as a promising direction for future research.

---

> > ### Comment · Reviewer_LKbx · 2025-08-05
> >
> > The concerns raised have been fully addressed for me.

---

> ### Author Response · Authors · 2025-08-05
>
> Thank you again for your constructive review and strong support of our work! We are glad that we have fully addressed your concerns. We greatly appreciate your valuable suggestions and will incorporate the discussions into the final version.

---

### Comment · Area_Chair_SAs1 · 2025-08-04
**Please carefully read the rebuttal and start the discussion**

Dear Reviewers and Authors,

Thank you all for your efforts so far. As the author–reviewer discussion period will conclude on **August 6**, please start the discussion as soon as possible.


**For Reviewers:**
Please read the authors’ responses and, if necessary, continue the discussion with them.

* If your concerns have been addressed, consider updating your review and score accordingly.

* If some concerns remain, or if you share concerns raised by other reviewers, clearly state these in your review and consider adjusting your review (positively or negatively).

* If you feel that your concerns have not been addressed, you may also choose to keep your review as is.

* I will follow up with you again during the reviewer–AC discussion period (August 7–13) to finalize the reviews and scores.


**For Authors:**
If you have not already done so, please respond to all questions raised by the reviewers. Keep your responses factual, concise, and ensure that every point raised is addressed.

Best regards,

The AC

---

### Decision · Program_Chairs · 2025-09-17

**Decision:**

Accept (poster)

**Comment:**

This paper presents REED (Representation-Enhanced Elucidation of Diffusion), a systematic framework for incorporating representation guidance into diffusion models. The authors provide theoretical insights into how high-quality pretrained representations can improve diffusion model training, and propose two strategies: (1) integrating multimodal representations with synthetic data, and (2) designing an optimal curriculum that balances representation learning with data generation. The framework is validated across three domains—image generation, protein sequence generation, and molecule generation—showing strong performance and accelerated training compared to existing methods.

Most reviewer concerns were effectively addressed during the rebuttal, and all reviewers agree that this is a significant contribution. The authors are encouraged to incorporate the provided feedback into the final revision.